# TRACE: TEMPORAL GROUNDING VIDEO LLM VIA CAUSAL EVENT MODELING

**Yongxin Guo**[1]    **Jingyu Liu**[2]    **Mingda Li**[2]    **Qingbin Liu**[2]    **Xi Chen**[2,*]    **Xiaoying Tang**[1,3,4,*]

[1]School of Science and Engineering, The Chinese University of Hong Kong, Shenzhen 518172, China
[2]Tencent PCG
[3]Shenzhen Institute of Artificial Intelligence and Robotics for Society (AIRS), Shenzhen, China
[4]Guangdong Provincial Key Laboratory of Future Networks of Intelligence, Shenzhen, China

## ABSTRACT

Video Temporal Grounding (VTG) is a crucial capability for video understanding models and plays a vital role in downstream tasks such as video browsing and editing. To effectively handle various tasks simultaneously and enable zero-shot prediction, there is a growing trend in employing video LLMs for VTG tasks. However, current video LLM-based methods rely exclusively on natural language generation, lacking the ability to model the clear structure inherent in videos, which restricts their effectiveness in tackling VTG tasks. To address this issue, this paper first formally introduces causal event modeling framework, which represents video LLM outputs as sequences of events, and predict the current event using previous events, video inputs, and textural instructions. Each event consists of three components: timestamps, salient scores, and textual captions. We then propose a novel task-interleaved video LLM called TRACE to effectively implement the causal event modeling framework in practice. The TRACEprocess visual frames, timestamps, salient scores, and text as distinct tasks, employing various encoders and decoding heads for each. Task tokens are arranged in an interleaved sequence according to the causal event modeling framework's formulation. Extensive experiments on various VTG tasks and datasets demonstrate the superior performance of TRACE compared to state-of-the-art video LLMs. Our model and code are avaliable at https://github.com/gyxxyg/TRACE.

## 1 INTRODUCTION

Video Temporal Grounding (VTG) is an important ability for video understanding models (Lin et al., 2023b), and has becoming the base of a series of downstream tasks like moment retrieval (Fabian Caba Heilbron & Niebles, 2015; Gao et al., 2017; Oncescu et al., 2021), dense video caption (Zhou et al., 2018; Tang et al., 2019), video highlight detection (Lei et al.; Liu et al., 2022), and video summarization (Song et al., 2015; Gygli et al., 2014). While non-generative models excel in moment retrieval and video highlight detection (Lei et al., 2021; Han et al., 2024; Wang et al., 2024a), they are inflexible, task-specific, and demand substantial fine-tuning for optimal performance. To tackle these challenges, recent research employs video LLMs as versatile models, integrating timestamp information into visual inputs, and fine-tuning them on VTG tasks (Ren et al., 2023; Huang et al., 2023; Wang et al., 2024b; Qian et al., 2024; Wang et al., 2024c; Wu et al., 2024) to enhance their performance and facilitate zero-shot prediction.

**Challenges posed by videos' inherent structures.** Despite reflecting human intent, current video LLM based approaches rely on pure natural language generation. As illustrated in Figure 1(a), this approach lacks a clear structure and indiscriminately blends information, such as timestamps and text captions. In contrast, videos possess an inherent structure that transcends mere textual description. To accurately describe or reason from a video, it is insufficient to rely solely on natural language text. Instead, the corresponding timestamps and salient scores are also essential components. Together, these elements provide a more comprehensive and structured understanding of the video content. Consequently, the gap between videos' structure and current video LLMs undermines the

---

*Corresponding authors.

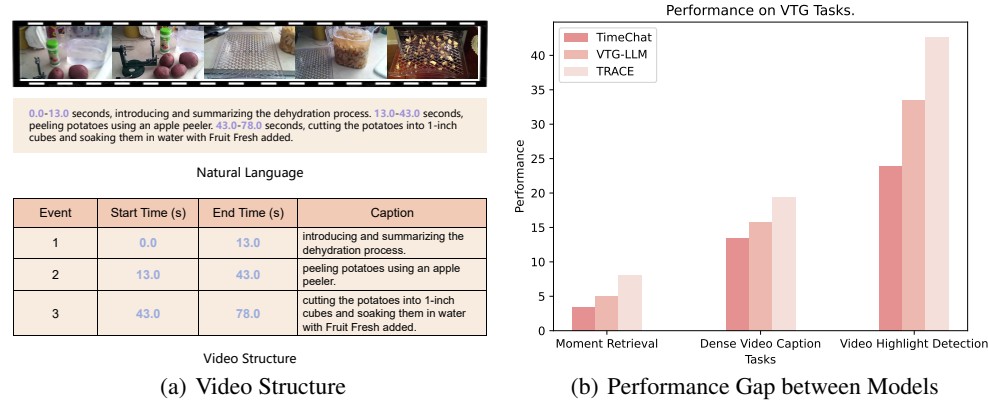

(a) Video Structure      (b) Performance Gap between Models

Figure 1: **Challenges posed by videos' inherent structures.** Figure 1(a) shows the difference between natural language and video structure, while Figure 1(b) highlights the performance gap between SOTA video LLMs (Ren et al., 2023; Guo et al., 2024) and TRACE. We present zero-shot performance results for video LLM approaches. Specifically, we report the performance of models using the CIDEr metric for the dense video captioning task on the Youcook2 dataset, $R@1_{IOU=0.7}$ for the moment retrieval task on the Charades-STA dataset, and HIT@1 for the highlight detection task on the QVHighlights dataset.

ability of video LLMs to effectively model video events, potentially making video LLMs difficult to achieve satisfactory results (Figure 1(b)) on VTG tasks.

**Causal event modeling as a solution.** In this paper, our primary goal is to develop a novel video LLM approach for resolving the mismatch between language modeling of LLMs and videos' inherent structure. Specifically, we concentrate on tackling two main challenges: (1) developing a theoretical framework that shifts from causal language modeling to structured event-based modeling, and (2) constructing a practical video LLM based on the theoretical framework to provide an effective solution. To accomplish this, we first introduce the causal event modeling framework, where video LLM outputs are represented as sequences of events, each containing timestamps, salient scores, and textual captions. The next events are predicted based on video inputs, text instructions, and preceding events. To effectively implement the causal event modeling framework in practice, we present a novel task-interleaved video LLM, TempoRAl grounding via Causal Event modeling (TRACE), as illustrated in Figure 2. The TRACE treats visual frames, timestamps, salient scores, and text as separate tasks, utilizing diverse encoders and decoding heads for each task, with task tokens sequenced in an interleaved manner. Furthermore, we develop an adaptive head-switching method for improved generation. Our numerical results across various VTG tasks reveal the superior performance of TRACE in comparison to state-of-the-art (SOTA) video LLMs.

**Our key contributions are summarized as follows:**

- We model the videos by a series of events, and propose causal event modeling framework to capture videos' inherent structure. We then present a novel task-interleaved video LLM model, TRACE, tailored to implement the causal event modeling framework through the sequential encoding/decoding of timestamps, salient scores, and textual captions.
- We conduct comprehensive experiments on multiple VTG tasks and datasets to verify the effectiveness of TRACE. The results reveal significant improvements of TRACE in comparison to SOTA video LLMs. Notably, TRACE improves zero-shot performance by 3.1 and 4.9% on Youcook2 (CIDEr and F1 Score), by 6.5% and 3.7% in Recall (IOU = {0.5, 0.7}) on Charades-STA, and by 10.3% and 9.2% for mAP and HIT@1 on QVHighlights. Moreover, surpassing existing video LLMs, TRACE achieves comparable performance to traditional non-generative and task-specific methods after fine-tuning, highlighting the potential of video LLMs to excel in VTG tasks.

## 2 RELATED WORKS

**Video temporal grounding.** Video Temporal Grounding (VTG) tasks aim to precisely identify the timestamps of events within a given video (Lin et al., 2023b). This includes various tasks such as

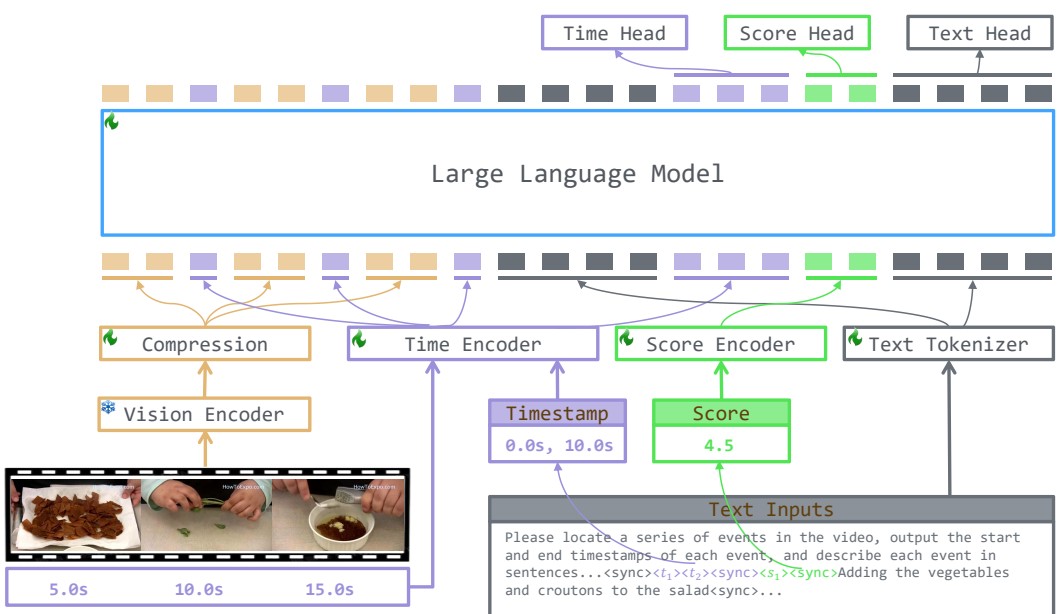

Figure 2: **Overview of the training process of TRACE model.** We employ a variety of encoders and heads to handle time, score, and text inputs and outputs. The timestamps of the sampled frames are converted into time tokens and subsequently integrated into the visual tokens of each frame. In the answer section, time tokens, score tokens, and text tokens are inserted in a sequential manner. The generation process of TRACE is summarized in Figure 4.

moment retrieval (Gao et al., 2017; Zala et al., 2023b; Oncescu et al., 2021; Hendricks et al., 2018a; Boris et al., 2024), dense video caption (Zellers et al., 2021; Zala et al., 2023b; Tang et al., 2019; Fabian Caba Heilbron & Niebles, 2015; Kim et al., 2024), video summarization (Song et al., 2015; Gygli et al., 2014; Hua et al., 2024), and video highlight detection (Lei et al., 2021; Xiao et al., 2023). For tasks such as moment retrieval, video summarization, and video highlight detection, traditional approaches primarily use large-scale video-text pre-training (Xu et al., 2021; Wang et al., 2022; Yan et al., 2022; Li et al., 2023d; Chen et al., 2024b; Tong et al., 2022; Zhao et al., 2024). Subsequently, they fine-tune the pretrained models by incorporating task-specific prediction heads. While these methods have demonstrated satisfactory results, they are resource-intensive for pre-training, lack zero-shot capabilities, can only handle one specific task per model, and often require additional fine-tuning for numerous downstream tasks. For the dense video caption task, Vid2Seq employs special time tokens to represent timestamps (Yang et al., 2023). Some approaches integrate additional input information, such as text queries from training datasets (Kim et al., 2024), while other models utilize different decoding heads to decode timestamps and textual captions (Wang et al., 2021; 2023a) in parallel. However, these architectures are specifically designed for the dense video caption task, cannot be easily adapted to LLM structures to fully harness the capacity of pretrained LLMs, and also lack zero-shot capabilities.

**Video LLMs for video temporal grounding.** Large language models (LLMs) (Kaplan et al., 2020; Achiam et al., 2023; Touvron et al., 2023) have demonstrated significant potential in acquiring knowledge and addressing real-world challenges using a zero-shot approach. Recent research has focused on integrating knowledge from other modalities, such as vision (Liu et al., 2024; Li et al., 2023a) and audio (Ghosal et al., 2023), to bolster the capabilities of LLMs. Within the visual domain, video large language models (video LLMs) have emerged as a crucial research area (Lin et al., 2023a; Maaz et al., 2023; Zhu et al., 2023; Song et al., 2024a;b). Traditional video LLMs (Zhang et al., 2023; Lin et al., 2023a; Li et al., 2023b; 2024; Cheng et al., 2024; Yao et al., 2024) have made considerable performance improvements in tasks such as video question answering, reasoning, and video captioning. However, these methods encounter difficulties in precisely pinpointing event timestamps within videos. To address this issue, TimeChat (Ren et al., 2023), VTimeLLM (Huang et al., 2023), and Hawkeye (Wang et al., 2024b) have attempted to overcome this limitation by

fine-tuning the video LLMs on VTG datasets. More recently, LITA (Huang et al., 2024) introduces fast-slow visual tokens and incorporates time tokens into LLM tokenizers. Momentor (Qian et al., 2024) suggests a time encoder to address time token quantization errors. VTG-LLM (Guo et al., 2024) integrates special time tokens and time position embeddings to improve the ability of video LLMs in comprehending timestamps. However, these methods do not take into account the inherent structure of videos and still cannot achieve satisfactory performance. In this paper, we propose the causal event modeling framework to provide structured video LLM outputs and design the TRACE model to address the proposed framework. Numerical results demonstrate significant performance gains of TRACE over existing video LLMs on VTG tasks.

## 3 TRACE

In this section, we aim to develop a novel video LLM that aligns well with video structures, addressing two questions: (1) how to model the structured video LLM outputs that are align well with video structures, and (2) how to implement theoretical models. We start by proposing *causal event modeling* framework to tackle "how to model". Then, we introduce TRACE to address "how to implement". We have included a detailed discussion about our framework in Appendix C.

### 3.1 MODELING THE INHERENT STRUCTURES OF VIDEOS

**Formulating outputs of video LLMs by events.** Given the instruction $\mathbf{I}$ and video visual inputs $\mathbf{F}$, we represent the outputs of video LLMs $\mathbf{R}$ as a series of events $\{e_1, e_2, \cdots, e_K\}$, with each event $e_k = (t_k, s_k, c_k)$ encompassing timestamps $t_k$, salient scores $s_k$, and textual captions $c_k$. In summary, we have

$$\mathbf{R} = \{e_1, e_2, \cdots, e_K\} = \{(t_k, s_k, c_k)|1 \le k \le K\}. \tag{1}$$

**Causal event modeling framework.** To effectively utilize the knowledge of pretrained LLMs, the design of causal event modeling shares the underlying intuition of causal language modeling, as formulated in the subsequent equation [1].

$$
\begin{aligned}
\mathcal{P}(e_k|e_{1:k-1}, \mathbf{I}, \mathbf{F}) &= \mathcal{P}(t_k, s_k, c_k|e_{1:k-1}, \mathbf{I}, \mathbf{F}), \\
&= \mathcal{P}(t_k|e_{1:k-1}, \mathbf{I}, \mathbf{F})\mathcal{P}(s_k|t_k, e_{1:k-1}, \mathbf{I}, \mathbf{F})\mathcal{P}(c_k|s_k, t_k, e_{1:k-1}, \mathbf{I}, \mathbf{F}),
\end{aligned} \tag{2}
$$

The next event $e_k$ is determined by textural instructions, visual inputs, and previous events. We can find that causal event modeling framework aligns well with the video structure (Figure 1(a)): (1) timestamps, salient scores, and textual captions are sequentially decoded within each event; (2) events are then ordered by timestamps.

### 3.2 TRACE: TASK-INTERLEAVED TEMPORAL GROUNDING VIDEO LLM

In Eq. 2, we introduce a formal causal event modeling framework to tackle the challenge of modeling structured video LLM outputs. This section illustrates the design of TRACE to implement the causal event modeling framework (Figure 2).

**Overview of TRACE.** As illustrated in Eq. 2, the causal event modeling framework necessitates encoding/decoding of visual frames ($\mathbf{F}$), text ($\mathbf{I}$ and $c_k$), timestamps ($t_k$), and scores ($s_k$). Consequently, the TRACE considers these elements as distinct tasks and employs the following design to efficiently manage them.

- *Separated multi-task processing.* TRACE utilizes separate encoders and decoding heads for each task to convert task inputs into task tokens and decode task tokens back to outputs (Sec. 3.2.1).
- *Task-interleaved sequence modeling.* Task tokens are sequenced in an interleaved manner according to Eq. 2 in TRACE and fed into LLM backbones (Sec. 3.2.2).
- *Adaptive head-switching mechanism for generation.* During generation, we implement an adaptive head-switching mechanism to select the appropriate decoding head for producing the next token (Sec. 3.2.3).

---

[1] Theoretically, the order of time, score, and text will not impact the results. We select one order here.

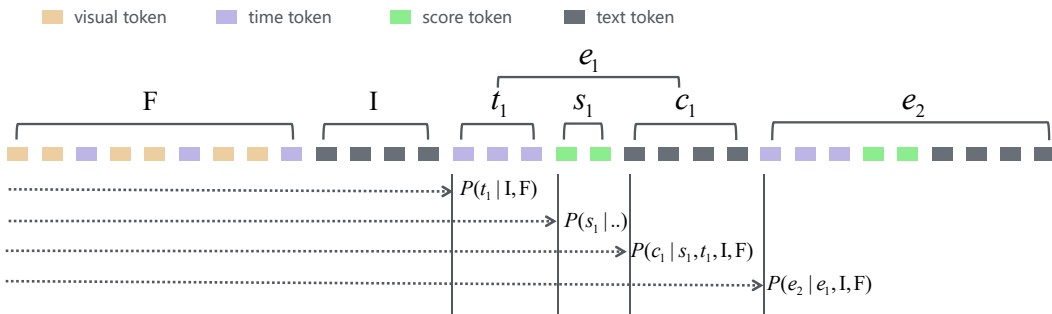

Figure 3: **Illustration of token sequence of TRACE.** Following Eq 2, the sequence begins with visual frame tokens (**F**) followed by instruction tokens (**I**). Event ($e$) tokens are structured in the sequence of time tokens ($t$), score tokens $s$, and text tokens $c$, with events ordered chronologically based on their occurrence time.

### 3.2.1 SEPARATED MULTI-TASK PROCESSING

TRACE consists of four unique tasks: visual frames, text, timestamps, and scores. Regarding text, we directly utilize the text tokenizer and LLM head of the LLM backbone (Mistral-7B-v0.2 (Jiang et al., 2023)). Moreover, we added a special token $\langle sync \rangle$ for indicating the end of text tasks. The processing for the other tasks is detailed below.

**Timestamps and scores processing.** For processing timestamps and score information, we employ two separate encoders and decoding heads, both of which share the same architecture. Specifically, each encoder is initialized with a tokenizer containing 13 tokens: 11 number tokens $\langle 0 \rangle, \cdots, \langle 9 \rangle, \langle . \rangle$ for representing timestamps/scores, $\langle sep \rangle$ to mark the end of each timestamp/score, and $\langle sync \rangle$ to signify the end of the current task. Token embeddings are initialized using LLM token embeddings.

In accordance with the research in VTG-LLM (Guo et al., 2024), we format each timestamp/score to the same length, comprising 4 whole-number parts, 1 dot, and 1 fractional part [2]. Subsequently, $\langle sep \rangle$ is inserted between timestamps/scores, and $\langle sync \rangle$ is appended at the end of each timestamp/score input sequence. For instance, the timestamp inputs [10.23, 125.37] will be tokenized into the following sequence: $\langle 0 \rangle\langle 0 \rangle\langle 1 \rangle\langle 0 \rangle\langle . \rangle\langle 2 \rangle\langle sep \rangle\langle 0 \rangle\langle 1 \rangle\langle 2 \rangle\langle 5 \rangle\langle . \rangle\langle 4 \rangle\langle sync \rangle$.

**Visual frames processing.** Given a $T$-frame video, we initially encode the frames using the pre-trained CLIP ViT-L (Radford et al., 2021), with each frame being encoded into 576 visual tokens. Subsequently, we employ Slot-Based Compression (Guo et al., 2024) to reduce the number of visual tokens to 8 per frame. Moreover, to integrate temporal information into the visual inputs, we use a time encoder to encode the timestamps of each sampled frame and remove the $\langle sync \rangle$ and $\langle sep \rangle$ tokens, resulting in 6 time tokens for each frame. Finally, we concatenate the 8 visual tokens with the 6 time tokens to form the visual inputs for each frame.

### 3.2.2 TASK-INTERLEAVED SEQUENCE MODELING

Utilizing the processed task tokens, we construct the sequence following Eq. 2. The token sequence order is illustrated in Figure 3.

**Inter-event sequence order.** The sequence commences with visual frame tokens **F** followed by textual instruction tokens **I**. For the events section, event tokens are sequenced according to the events' occurrence time to align with the causal event modeling formula $\mathcal{P}(e_k|e_{1:k-1}, \mathbf{I}, \mathbf{F})$.

**Intra-event sequence order.** For each event, in accordance with Eq. 2, tokens are arranged sequentially by time tokens ($\mathcal{P}(t_k|e_{1:k-1}, \mathbf{I}, \mathbf{F})$), score tokens ($\mathcal{P}(s_k|t_k, e_{1:k-1}, \mathbf{I}, \mathbf{F})$), and text tokens ($\mathcal{P}(c_k|s_k, t_k, e_{1:k-1}, \mathbf{I}, \mathbf{F})$). Consequently, the causal event modeling framework (Eq. 2) emerges as a specialized autoaggressive model, featuring a unique sequence order that closely aligns with video structures.

---

[2]Different from timestamps, scores will be encoded to 3 score tokens, including 1 whole-number parts, 1 dot, and 1 fractional part.

Table 1: **Datasets used for TRACE training process.** "Compressed" indicates that datasets are condensed by retaining only one sample for samples with identical videos but varying instructions.

| Stage | Datasets | Quantity |
|---|---|---|
| Stage 1 | Valley, LLaVA_Image, TextVR, ShareGPT4Video, VTG-IT | 1.9M |
| Stage 2 | Valley (Compressed), TextVR (Compressed), ShareGPT4Video (Compressed), VTG-IT, ActivityNet Captions, VideoChatGPT, InternVid, Next-QA | 0.9M |

### 3.2.3 ADAPTIVE HEAD-SWITCHING MECHANISM FOR GENERATION

**Using $\langle sync \rangle$ token for adaptive head switching.** Since TRACE employs distinct decoding heads for various tasks during training, selecting the appropriate decoding head during generation based on previously decoded tokens is crucial. This selection is facilitated by the $\langle sync \rangle$ token. As illustrated in Figure 4, TRACE generates tokens in the sequence of time, score, and text tokens. Detection of the $\langle sync \rangle$ token prompts TRACE to switch decoding heads accordingly. The heads are cycled switched in the order of time head - score head - text head.

### 3.3 TRAINING STRATEGY AND DATA PREPARATION

This section outlines the TRACE training process, which includes two stages. For the stage 1, task modules such as the vision compression layer, task encoder, and task heads are trained for initialization. For the stage 2, the LLM backbone is fine-tuned while keeping the task modules tuned. Detailed settings and datasets are presented below. Due to the page limitation, detailed annotation examples for each task, and details about data filtering and processing are provided in Appendix A.

**Stage 1: Initialization of task modules.** In stage 1, task modules such as the vision compression layer, time encoder/head, score encoder/head, and text tokenizer/head are trained while the vision encoder and LLM backbone remain fixed. As shown in Table 1, stage 1 primarily utilizes two groups of datasets.

- *Image and video caption datasets for initializing the visual compression layer.* This group of datasets including Valley (Luo et al., 2023b), LLaVA_Image (Liu et al., 2024), TextVR (Wu et al., 2025), and a randomly sampled subset of ShareGPT4Video (Chen et al., 2024a) datasets.
- *VTG datasets for task encoder/head initialization.* We use VTG-IT dataset in this group.

For stage 1 training, we uniformly sample 128 frames from each video. The learning rate is set to 1e-3, and models are trained for one epoch with a batch size of 128.

**Stage 2: Instruction tuning for enhancing VTG capacity.** In Stage 2, the LLM backbone and task modules are fine-tuned, with only the vision encoder remaining fixed. As shown in Table 1, stage 2 primarily utilizes three groups of datasets.

- *VTG instruction tuning datasets for enhancing VTG capacity.* We use VTG-IT (Guo et al., 2024), ActivityNet Captions (Fabian Caba Heilbron & Niebles, 2015), and a subset of InternVid (Wang et al., 2023b), resulting in a total of 635K data samples. Low-quality samples were filtered out, and the VTG-IT-VHD and VTG-IT-VS datasets were re-annotated. Additional details can be found in Appendix A.
- *Video caption datasets for maintaining the quality of the visual compression layers.* We use parts of the video data from stage 1, such as Valley (Luo et al., 2023b), TextVR (Wu et al., 2025), and ShareGPT4Video (Chen et al., 2024a) datasets. These datasets are compressed by retaining only one sample for samples with identical videos but different instructions, yielding 284K data.
- *Video question answering datasets to enhance TRACE's reasoning capabilities.* We use VideoChatGPT (Maaz et al., 2023) and Next-QA (Xiao et al., 2021) in this part.

For each video, the content is uniformly divided into 128 clips, with one frame randomly sampled from each clip. The learning rate is set to 5e-6, and the models are trained for two epochs using a batch size of 128.

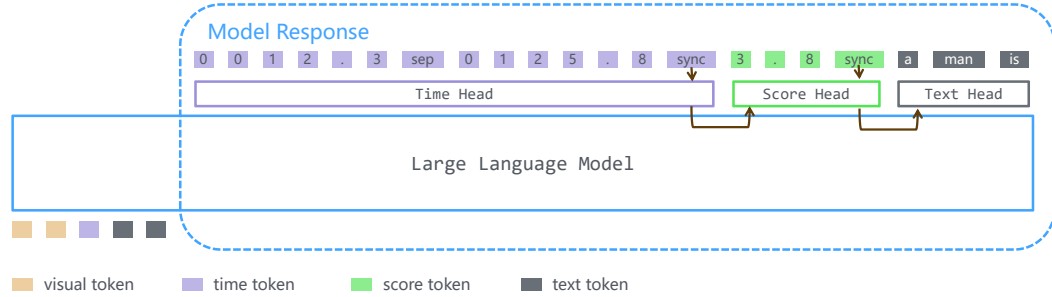

Figure 4: **Generation process of TRACE.** The TRACE generate tokens following the order of time tokens, score tokens, and text tokens. The decoding heads are switched when $\langle sync \rangle$ tokens are generated.

## 4 EXPERIMENTS

Detailed experimental settings and hyper-parameters can be found in Appendix B.1. Numerical results on more video understanding benchmarks and more ablation studies can be found in Appendix B.2. Case studies can be found in Appendix B.3.

### 4.1 EVALUATION DATASETS, METRICS, AND BASELINE MODELS.

We evaluate the model performance on three different tasks:

- *Dense video caption.* We use Youcook2 (Zhou et al., 2018) and ActivityNet Captions (Fabian Caba Heilbron & Niebles, 2015) datasets as the evaluation datasets. The evaluation metrics include CIDEr (Vedantam et al., 2015), METEOR (Banerjee & Lavie, 2005), and SODA_c (Fujita et al., 2020) for assessing the quality of the captions. These metrics are averaged under different IoU thresholds $\{0.3, 0.5, 0.7, 0.9\}$, following previous studies (Ren et al., 2023; Huang et al., 2023). Additionally, we report the F1 score to measure the model's ability to accurately locate timestamps.
- *Moment retrieval.* We utilize test set of Charades-STA (Gao et al., 2017) for the moment retrieval task and report the recall at IOU thresholds of 0.5 and 0.7. Additionally, we present the mIOU results.
- *Video highlight detection.* We employ the validation set of the QVHighlights dataset (Lei et al., 2021) and report the mean average precision (mAP) with IOU thresholds of 0.5 and 0.75, as well as the HIT@1, which represents the hit ratio of the highest scored clip.

For baseline models, we select Valley (Luo et al., 2023b), VideoChat (Li et al., 2023b), Video-ChatGPT (Maaz et al., 2023), and Video-LLaMA (Zhang et al., 2023) as examples of traditional video LLMs. For video LLMs specifically designed for VTG tasks, we choose TimeChat (Ren et al., 2023), VTimeLLM (Huang et al., 2023), Momentor (Qian et al., 2024), HawkEye (Wang et al., 2024b), and VTG-LLM (Guo et al., 2024).

### 4.2 PERFORMANCE OF TRACE

**Superior zero-shot performance of TRACE over other video LLMs.** In Table 2, we show the zero-shot performance of TRACE compare to SOTA video LLM baselines. The results show that

- **Suprior zero-shot performance.** As shown in Table 2, TRACE significantly outperforms other video LLMs by a substantial margin across all three datasets. Notably, it achieves a 3.1 and 4.9% performance improvement on the Youcook2 dataset using the CIDEr and F1 Score metrics; a 6.5% and 3.7% performance increase in Recall with IOU = $\{0.5, 0.7\}$ thresholds on the Charades-STA dataset; and a 10.3% and 9.2% performance gain for the mAP and HIT@1 metrics on the QVHighlights dataset.
- **Better performance than task-specific models and larger LLMs.** As shown in Table 2, as a generalist model capable of handling various tasks, the performance of TRACE surpasses that of task-specific models like HawkEye (Wang et al., 2024b). Furthermore, the 7B TRACE model outperforms the VTimeLLM (13B) model (Huang et al., 2023), further validating the advantages of the TRACE architecture.

Table 2: **Zero-shot performance of algorithms over various tasks.** We evaluated the performance of TRACE using the Youcook2, Charades-STA, and QVHighlights datasets. We highlight the best results for each block using **bold font**. The Valley, VideoChat-Embed, and Video-LLaMA results are elaborated from previous studies (Ren et al., 2023; Huang et al., 2023; Qian et al., 2024). The results with transparent text indicates unfair comparison (13B).

| Model | Youcook2 | | | Charades-STA | | QVHighlights | |
|---|---|---|---|---|---|---|---|
| | SODA_c | CIDEr | F1 Score | R@1$_{(IOU=0.5)}$ | R@1$_{(IOU=0.7)}$ | mAP | HIT@1 |
| *Traditional Video LLMs* | | | | | | | |
| Valley (7B) | 0.1 | 0.0 | 1.5 | 4.7 | 1.6 | 10.9 | 15.2 |
| VideoChat (7B) | 0.2 | 0.6 | 3.4 | 3.2 | 1.4 | 13.1 | 18.1 |
| Video-LLaMA (7B) | 0.0 | 0.0 | 0.1 | 2.7 | 1.2 | 11.3 | 15.6 |
| *Temporal Grounding Video LLMs* | | | | | | | |
| TimeChat (7B) | 1.2 | 3.4 | 12.6 | 32.2 | 13.4 | 14.5 | 23.9 |
| VTimeLLM (7B) | | | | 27.5 | 11.4 | | |
| VTimeLLM (13B) | | | | 34.3 | 14.7 | | |
| Momentor (7B) | | | | 26.6 | 11.6 | 7.6 | |
| HawkEye (7B) | | | | 31.4 | 14.5 | | |
| VTG-LLM (7B) | 1.5 | 5.0 | 17.5 | 33.8 | 15.7 | 16.5 | 33.5 |
| TRACE (7B) | **2.2** | **8.1** | **22.4** | **40.3** | **19.4** | **26.8** | **42.7** |

**Performance of TRACE on ActivityNet Captions dataset.** In Table 4, we show the performance of TRACE on ActivityNet Captions dataset. All the reported algorithms except for Momentor (Qian et al., 2024) have incorporated the ActivityNet Captions dataset as part of the training data. Results show that the TRACE attains the best performance in moment retrieval tasks and demonstrates comparable results to VTimeLLM in dense video caption tasks.

## 4.3 ABLATION STUDIES OF TRACE.

**The causal event modeling framework enhances model performance in VTG tasks.** In the 'Ablation Studies on Architecture' section of Table 3, we conducted experiments without utilizing the causal event modeling framework. The results indicate that employing the causal event modeling framework significantly improves model performance, and TRACE can achieve better results even when sampling fewer video frames.

**Using different encoders and decoding heads for different tasks is essential for TRACE to achieve the best result.** In the "w/o independent ecoder/heads" part of Table 3, we performed ablation studies by not utilizing separate encoders and decoder heads for different tasks. Instead, we directly incorporated time tokens and score tokens into the text tokenizers. The results suggest that using shared encoder/decoding heads for causal event modeling framework significantly disrupts the prelearned knowledge of LLMs, leading to irrelevant and meaningless responses.

**The performance of TRACE improves with the increase in the number of frames.** We conducted ablation studies on the number of sampled frames, as presented in Table 3. The results show that (1) the performance of TRACE enhances as the number of sampled frames increases; (2) the performance of TRACE is comparable or even superior to SOTA video LLMs like VTG-LLM and TimeChat when sampling just 8 frames, demonstrating the effectiveness of the TRACE model architecture.

**Incorporating InternVid (Wang et al., 2023b) and ActivityNet Captions (Fabian Caba Heilbron & Niebles, 2015) datasets boost TRACE performance on long videos.** As illustrated in Figure 5, we carried out ablation studies by exclusively using VTG-IT as the training data for VTG tasks. The results indicate that the performance of TRACE on long videos improves when incorporating internVid and ActivityNet Captions datasets, leading to enhanced performance on Youcook2, QVHighlights, and ActivityNet Captions datasets. Conversely, the performance of TRACE on short videos slightly decreases (Charades-STA), suggesting that the annotations in the internVid and ActivityNet Captions datasets may not be as accurate as those in short video annotations.

Table 3: **Ablation studies of TRACE.** All the algorithms solely utilize VTG-IT (Guo et al., 2024) during fine-tuning for efficient evaluation. The "w/o causal event modeling " approach indicates the use of natural language-style inputs similar to previous studies (Guo et al., 2024; Ren et al., 2023). The "w/o independent encoder/heads" approach signifies directly adding new tokens to the LLM tokenizer instead of employing separate encoders/heads for different tasks. We highlight the best results using **bold font** for each block.

| Models | Frame Number | Youcook2 | | | Charades-STA | |
|---|---|---|---|---|---|---|
| | | SODA_c | CIDEr | F1 Score | R@1$_{(IOU=0.5)}$ | R@1$_{(IOU=0.7)}$ |
| *Ablation Studies on Architecture* | | | | | | |
| w/o causal event modeling | 96 | 1.4 | 4.3 | 17.2 | 29.7 | 14.0 |
| w/o independent encoder/heads | 64 | —-Fail to Follow Instruction—- | | | | |
| TRACE (VTG-IT) | 64 | **1.9** | **6.9** | **21.4** | **37.0** | **17.0** |
| *Ablation Studies on Frame Number* | | | | | | |
| TRACE (VTG-IT) | 8 | 1.4 | 5.0 | 18.6 | 28.8 | 13.6 |
| TRACE (VTG-IT) | 64 | 1.9 | 6.9 | **21.4** | 37.0 | 17.0 |
| TRACE (VTG-IT) | 128 | **2.1** | **7.5** | **21.4** | **41.2** | **20.0** |

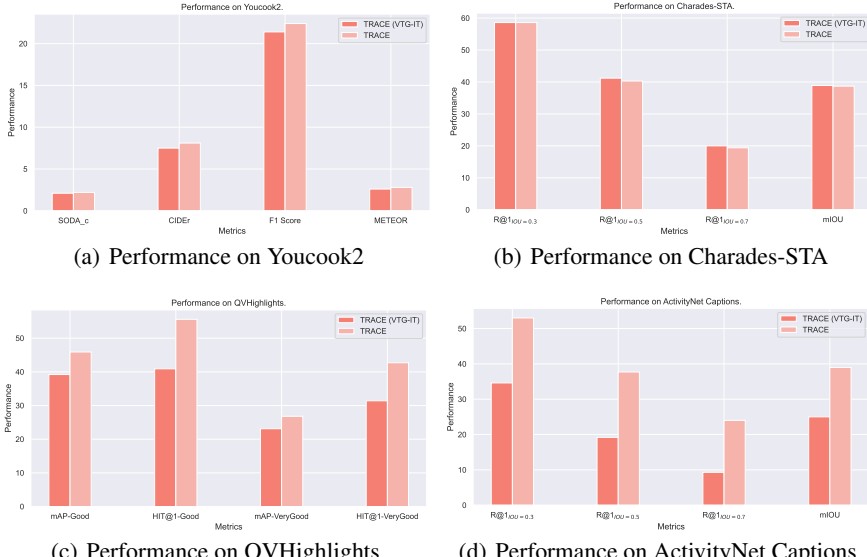

(a) Performance on Youcook2        (b) Performance on Charades-STA

(c) Performance on QVHighlights        (d) Performance on ActivityNet Captions

Figure 5: **Ablation studies on data utilized while training TRACE.** We conduct experiments solely utilizing VTG-IT and compare its performance with that of the original TRACE.

### 4.4 FINE-TUNED PERFORMANCE OF TRACE.

**Competitive performance of TRACE to traditional methods after fine-tuning.** In Table 5, we fine-tune the TRACE for 3 epochs on Youcook2 and Charades-STA datasets [3]. The results indicate that

- *TRACE significantly outperform generalist baselines.* In contrast to TimeChat and VTG-LLM, which struggle to attain satisfactory performance even after fine-tuning, the TRACE derives significant benefits from fine-tuning and achieves notably better performance than generalist baselines. These results further substantiate that our enhancements to the model architecture are crucial for VTG tasks.
- *TRACE achieve comparable performance to non-generative and task-specific SOTAs.* As depicted in Table 5, the TRACE achieves new SOTA results on Youcook2 (without audio inputs). Furthermore, the performance of TRACE on the Charades-STA dataset is also competitive with non-generative models such as InternVideo2 and VDI. *However, these methods cannot handle various tasks simultaneously and lack zero-shot capability – the contribution of TRACE herein.*

---

[3]Results on QVHighlights can be found in Appendix B.2.

Table 4: **Performance of TRACE on ActivityNet Captions dataset.** The evaluation of TimeChat's and VTG-LLM's results was conducted using the official provided checkpoints. The * indicates zero-shot evaluation. We highlight the best and the second best results using **bold** and underline.

| Models | Dense Video Caption | | | | Moment Retrieval | | |
|---|---|---|---|---|---|---|---|
| | METEOR | SODA_c | CIDEr | F1 Score | R@1$_{(IOU=0.5)}$ | R@1$_{(IOU=0.7)}$ | mIOU |
| VTimeLLM | **6.8** | 5.8 | **27.6** | | 29.5 | 14.2 | 31.4 |
| Momentor* | 4.7 | 2.3 | 14.9 | | 23.0 | 12.4 | 29.3 |
| TimeChat | 5.7 | 4.7 | 19.0 | 36.9 | 4.6 | 2.0 | 6.9 |
| VTG-LLM | 5.9 | 5.1 | 20.7 | 34.8 | 8.3 | 3.7 | 12.0 |
| TRACE | 6.4 | **6.0** | 25.9 | **39.3** | **37.7** | **24.0** | **39.0** |

Table 5: **Fine-tuned performance of TRACE.** We fine-tune the TRACE for 3 epochs on the Youcook2 and Charades-STA datasets. We emphasize the best and second best results using **bold font** and underline. For Youcook2 dataset, we choose Vid2Seq (Yang et al., 2023), PDVC (Wang et al., 2021), and CM$^2$ (Kim et al., 2024) as task-specific baselines. The results depicted in gray indicate unfair comparisons due to additional audio inputs and different architectures. For charades-STA dataset, we choose InternVideo2-6B (Wang et al., 2024a), VDI (Luo et al., 2023a), and Moment-DETR (Lei et al., 2021) as examples of non-generative models.

| Model | Youcook2 | | | Model | Charades-STA | |
|---|---|---|---|---|---|---|
| | SODA_c | CIDEr | F1 Score | | R@1$_{(IOU=0.5)}$ | R@1$_{(IOU=0.7)}$ |
| *Task-Specific Models* | | | | *Non-Generative Models* | | |
| PDVC | 4.4 | 22.7 | | InternVideo2-6B | 70.0 | 49.0 |
| Vid2Seq (Audio Input) | 7.9 | 47.1 | 27.3 | VDI | 52.3 | 31.4 |
| Vid2Seq | 5.7 | 25.3 | 23.5 | Moment-DETR | 55.7 | 34.2 |
| CM$^2$ | 5.3 | 31.7 | 28.4 | *Generative Models* | | |
| *Generalist Models* | | | | HawkEye | 58.3 | 28.8 |
| TimeChat | 3.4 | 11.0 | 19.5 | TimeChat | 46.7 | 23.7 |
| VTG-LLM | 3.6 | 13.4 | 20.6 | VTG-LLM | 57.2 | 33.4 |
| TRACE | **6.7** | **35.5** | **31.8** | TRACE | **61.7** | **41.4** |

## 5 Conclusion and Future Works

In this paper, our goal is to address the mismatch between video structure and video LLMs on VTG tasks, and propose a causal event modeling framework and the TRACE model as a solution. Numerical results indicate the superior zero-shot performance of TRACE compared to other video LLM baselines, and TRACE also achieves competitive performance relative to traditional non-generative and task-specific models after fine-tuning. By overcoming the inherent limitations of video LLM architectures, TRACE demonstrates the potential of video LLMs on VTG tasks, and we believe that the TRACE could be a strong foundation for future research on video LLMs in VTG tasks.

However, there are future works that can further enhance the capabilities of TRACE. For instance, TRACE relies on the pre-trained decoder-only LLMs, and only using previous events to predict the next event, which may not discover the complex event relationships as pointed out by previous studies (Yi et al., 2019; Girdhar & Ramanan, 2019; Li et al., 2020), As a remedy, we can use the outputs of causality discovery models (Liang et al., 2022; Chen et al., 2024c) as supplementary inputs for TRACE to provide a more comprehensive understanding of video contents. Furthermore, expanding the annotation of more video understanding tasks by incorporating the occurrence timestamps of QA pairs and the matching score between questions and answers could significantly improve the overall performance of TRACE.

## Acknowledgments

This work is supported in part by the funding from Shenzhen Institute of Artificial Intelligence and Robotics for Society, in part by the Shenzhen Key Lab of Crowd Intelligence Empowered Low-Carbon Energy Network (Grant No. ZDSYS202206061006010002), in part by Shenzhen Stability

Science Program 2023, and in part by the Guangdong Provincial Key Laboratory of Future Networks of Intelligence (Grant No. 2022B1212010001).

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

C​ONTENTS OF A​PPENDIX

## A  D​ATASET P​REPARATION

### A.1  D​ETAILS OF DATA FORMAT

In this section, we introduce the details of the data format we used while training TRACE. For the VTG datasets, in addition to ActivityNet Captions dataset and InternVid dataset, we directly use the annotation collected by VTG-IT (Guo et al., 2024). In detail, the annotations can be categories into the following four types:

- **General tasks (Figure 6).** For general tasks such as video captioning, image captioning, and video question answering, the answer component of the data does not include timestamps or scores. Consequently, we employ a single token $\langle sync \rangle$ as a placeholder for timestamps and scores, signifying an empty response for this part of response. This kind of annotation including LLaVA_Image (Liu et al., 2024), Valley (Luo et al., 2023b), TextVR (Wu et al., 2025), ShareGPT4Video (Chen et al., 2024a), VideoChatGPT (Maaz et al., 2023), and Next-QA (Xiao et al., 2021) datasets.

- **Dense video caption task (Figure 7).** The Dense Video Captioning task solely comprises timestamps and textual captions responses. As a result, we use a single token $\langle sync \rangle$ as a placeholder for scores. The datasets of this task include HiREST$_{step}$ (Zala et al., 2023a), COIN (Tang et al., 2019), ActivityNet Captions (Fabian Caba Heilbron & Niebles, 2015), VTG-IT-DVC (Guo et al., 2024), and InternVid (Wang et al., 2023b) datasets.

- **Moment retrieval task (Figure 8).** Similiar to dense video caption task, moment retrieval solely comprises timestamps and textual captions responses. The moment retrieval task including HiREST$_{grounding}$ (Zala et al., 2023a), QuerYD (Oncescu et al., 2021), DiDeMo (Hendricks et al., 2018b), VTG-IT-MR (Guo et al., 2024), and InternVid (Wang et al., 2023b) datasets.

- **Video highlight detection task (Figure 9).** For the video highlight detection task, we utilize the query as the textual response for all highlight moments. In this case, we employ the VTG-IT-VHD (Guo et al., 2024) dataset.

- **Video summarization task (Figure 10).** The video summarization task employs the caption of each event as the textual response. We use the VTG-IT-VS (Guo et al., 2024) dataset here.

### A.2  P​ROCESSING I​NTERN​V​ID

**Dense video caption data.**  The dense video caption data is constructed based on the InternVid-Full annotations. For each video, we opt not to use the annotation of that video as dense video caption data if any of the following checklist items are met:

- There exists a caption with fewer than 5 words.

- There exists a caption very similar to another caption within one video. We use fuzzywuzzy package here and set the threshold to 70.

```
{
        "id": "0",
        "video": "valley/076101_076150/1043215450.mp4",
        "conversations": [
            {
                "from": "human",
                "value": "<video>\nWrite a terse but informative summary of the following video
clip."
            },
            {
                "from": "gpt",
                "value": "<sync><time><score>Oaxaca de juarez, mexico - circa 1970: mexican
tourists on the square of the cathedral of our lady of the assumption in the city of oaxaca.
archival of mexico in oaxaca state in the 1970s."
            }
        ],
        "times": [
            []
        ],
        "scores": [
            []
        ]
    }
```

Figure 6: **Annotation example of video caption task.**

```
{
    'video': 'activitynet/videos/v_lIbRuIm9MxI.mp4',
    'id': 370958,
    'conversations': [
        {
            'from': 'human', 'value': '<video>\nScrutinize the video and determine multiple
occurrences, providing their initial and final timestamps as well as a summary of each action.'
        },
        {
            'from': 'gpt', 'value':
'<sync><time><time><time><time><time><time><time><time><time><time><time><time><time><time><score>h
owcast logo is on the
screen.<sync><time><time><time><time><time><time><time><time><time><time><time><time><time><time><s
core>two men are sitting in the middle of drums and explaining about the
instruments.<sync><time><time><time><time><time><time><time><time><time><time><time><time><time><ti
me><score>the men begin playing the
drums.<sync><time><time><time><time><time><time><time><time><time><time><time><time><time><time><sc
ore>the go back to talking about the drums.'
        }
    ],
    'scores': [
        [], [], [], []
    ],
    'times': [
        [0, 5.42], [5.42, 86.09], [86.77, 120.67], [120.67, 135.58]
    ]
}
```

Figure 7: **Annotation example of dense video caption task.**

```
{
    "video": "videochatgpt_tune/v_QOlSCBRmfWY.mp4",
    "id": 0,
    "conversations": [
        {
            "from": "human",
            "value": "<video>\nGive you a textual query: 'a young woman is seen standing in a room
and leads into her dancing.' When does the described content occur in the video? Please return the
timestamp in seconds."
        },
        {
            "from": "gpt",
            "value":
"<sync><time><time><time><time><time><time><time><time><time><time><time><time><time><time><score>a
young woman is seen standing in a room and leads into her dancing."
        }
    ],
    "scores": [
        []
    ],
    "times": [
        [0.83, 19.86]
    ]
}
```

Figure 8: **Annotation example of moment retrieval task.**

- The number of events is less than 5 or greater than 50.

- There are special characters present, excluding letters, spaces, and dots.

**Moment retrieval data.** We discovered that directly utilizing the InternVid-Full annotations to construct the moment retrieval data leads to suboptimal performance, likely due to the imprecise timestamp annotations. Consequently, we employ the InternVid-10M-FLT-INFO annotation to construct the moment retrieval data, which is a filtered subset of the InternVid-Full annotation provided by the authors.

## A.3 PROCESSING VTG-IT

In this section, we describe the processing of the VTG-IT dataset. For the dense video caption and moment retrieval tasks, we directly utilize the annotations provided by VTG-IT. However, the data for video highlight detection and video summarization tasks supplied by VTG-IT often have uniform salient scores within each video. As a result, our goal here is to enhance the quality of data for these two tasks, specifically video highlight detection and video summarization.

- **Video highlight detection task.** For each event in the dense video caption task, we initially divide each event into a maximum of 20 clips. Subsequently, we compute the similarity between the frames within each clip and the event captions using ViT-G/14 from EVA-CLIP (Sun et al., 2023). The similarity scores are then normalized to a Gaussian distribution to serve as the highlight score. In detail, the clips with scores higher than [2.275%, 3.593%, 5.480%, 8.076%, 11.507%, 15.866%, 21.186%, 27.425%, 34.458%, 42.074%, 50.000%, 57.926%, 65.542%, 72.575%, 78.814%, 84.134%, 88.493%, 91.924%, 94.520%, 96.407%, 97.725%] of other clips will be assign scores [1.0, 1.2, 1.4, 1.6, 1.8, 2.0, 2.2, 2.4, 2.6, 2.8, 3.0, 3.2, 3.4, 3.6, 3.8, 4.0, 4.2, 4.4, 4.6, 4.8, 5.0].

- **Video summarization task.** The data for the video summarization task is built upon the foundation of the video highlight detection task and the dense video caption task. Specifically, for each event in the dense video caption task data, we select the clip with the highest score to serve as the summarization clip for that particular event.

```
{
    "video": "qvhighlights/videos/train/j7rJstUseKg_210.0_360.0.mp4",
    "id": 1,
    "conversations": [
        {
            "from": "human",
            "value": "<video>\nIn the video presented, seek moments that are a perfect
match with 'Man in baseball cap eats before doing his interview.'. It's vital to notate
their timestamps and to score each based on their level of saliency."
        },
        {
            "from": "gpt",
            "value":
"<sync><time><time><time><time><time><time><time><score><score><score><score>Man in
baseball cap eats before doing his
interview.<sync><time><time><time><time><time><time><time><score><score><score><score>Ma
n in baseball cap eats before doing his
interview.<sync><time><time><time><time><time><time><time><score><score><score><score>Ma
n in baseball cap eats before doing his
interview.<sync><time><time><time><time><time><time><time><score><score><score><score>Ma
n in baseball cap eats before doing his
interview.<sync><time><time><time><time><time><time><time><score><score><score><score>Ma
n in baseball cap eats before doing his
interview.<sync><time><time><time><time><time><time><time><score><score><score><score>Ma
n in baseball cap eats before doing his
interview.<sync><time><time><time><time><time><time><time><score><score><score><score>Ma
n in baseball cap eats before doing his
interview.<sync><time><time><time><time><time><time><time><score><score><score><score>Ma
n in baseball cap eats before doing his
interview.<sync><time><time><time><time><time><time><time><score><score><score><score>Ma
n in baseball cap eats before doing his interview."
        }
    ],
    "scores": [
        [2.6666666666666665], [3.0], [2.0], [1.6666666666666667], [2.6666666666666665],
[2.3333333333333335], [2.0], [1.6666666666666667], [2.3333333333333335]
    ],
    "times": [
        [96], [98], [100], [102], [104], [106], [108], [110], [112]
    ]
}
```

Figure 9: **Annotation example of video highlight detection task.**

```
{
    'video': 'yttemporal/videos/video-mqtp71vzGrg',
    'id': 370616,
    'conversations': [
        {
            'from': 'human', 'value': '<video>\nSummarize the video by pinpointing key frames that
encompass the core storyline.'
        },
        {
            'from': 'gpt', 'value':
'<sync><time><time><time><time><time><time><time><score><score><score><score>Introduction and
materials needed for the DIY disco
ball.<sync><time><time><time><time><time><time><time><score><score><score><score>Cutting the CDs
into small pieces using scissors.'
        }
    ],
    'scores': [
        [5.0], [4.6]
    ],
    'times': [
        [34.534498828324], [51.38466492330334]
    ]
}
```

Figure 10: **Annotation example of video summarization task.**

Table 6: **Detailed training setting and hyper-parameters.**

| Setting | Stage 1 | Stage 2 |
|---|---|---|
| Computation | 16 ATN 910B | 16 ATN 910B |
| Vision Encoder | openai/clip-vit-large-patch14-336 | openai/clip-vit-large-patch14-336 |
| DeepSpeed Stage | Zero3 Offload | Zero3 Offload |
| LLM | Mistral-7B-v0.2 | Mistral-7B-v0.2 |
| Batch Size | 128 | 128 |
| Num Frames | 128 | 128 |
| Frame Sample | Uniform | Split to 128 clips then random within clip |
| Train Epochs | 1 | 2 |
| Learning Rate | 1e-3 | 5e-6 |
| LR Scheduler | Cosine | Cosine |
| Model Max Length | 4096 | 4096 |

Table 7: **Ablation studies on training data (Youcook2 and Charades-STA).**

| Models | Youcook2 | | | | Charades-STA | | |
|---|---|---|---|---|---|---|---|
| | SODA_c | CIDEr | F1 Score | METEOR | R@1$_{(IOU=0.5)}$ | R@1$_{(IOU=0.7)}$ | mIOU |
| TRACE (VTG-IT) | 2.1 | 7.5 | 21.4 | 2.6 | 41.2 | 20.0 | 38.9 |
| TRACE | 2.2 | 8.1 | 22.4 | 2.8 | 40.3 | 19.4 | 38.7 |
| TRACE-uni | 2.3 | 8.6 | 22.4 | 2.9 | 43.7 | 21.0 | 41.5 |

Table 8: **Ablation studies on training data (ActivityNet Captions).**

| Models | Dense Video Caption | | | | Moment Retrieval | | |
|---|---|---|---|---|---|---|---|
| | SODA_c | CIDEr | F1 Score | METEOR | R@1$_{(IOU=0.5)}$ | R@1$_{(IOU=0.7)}$ | mIOU |
| TRACE (VTG-IT) | 5.8 | 24.7 | 38.9 | 6.0 | 19.2 | 9.3 | 25.0 |
| TRACE | 6.0 | 25.9 | 39.3 | 6.4 | 37.7 | 24.0 | 39.0 |
| TRACE-uni | 6.4 | 29.2 | 40.4 | 6.9 | 38.2 | 24.7 | 39.4 |

# B EXPERIMENTS

## B.1 DETAILED EXPERIMENTAL SETTINGS

We report the detailed model architecture design and training hyper-parameters in Table 6. The training takes about 5 days for stage 1 and 5 days for stage 2.

## B.2 ADDITIONAL EXPERIMENT RESULTS

**Ablation studies on training data.** In Tables 7 and 8, we report the performance of TRACE using different training data. The TRACE-uni indicates incorporating additional general video understanding data from a subset of LLaVA-Video-178k (Zhang et al., 2024) (specifically the perceptiontest and YouTube parts). The results show that (1) the performance of TRACE on long videos increase while using the original TRACE setting. (2) The performance of TRACE slightly reduced on short videos (Charades-STA) while using the original TRACE setting; (3) Although not adding more VTG data, TRACE-uni outperforms trace in both VTG tasks and general video understanding tasks.

**Fine-tuned performance on QVHighlights.** In Table 9, we show the performance of TRACE on QVHighlights dataset after fine-tuning. The results indicate that TRACE significantly outperform other video LLMs by a large margin.

Table 9: **Fine-tuned performance of algorithms on QVHighlights datasets.** We fine-tune the Algorithm on QVHighlights datasets.

| Model | QVHighlights | |
|---|---|---|
| | mAP | HIT@1 |
| TimeChat | 21.7 | 37.9 |
| VTG-LLM | 24.1 | 41.3 |
| TRACE | **31.8** | **51.5** |

Table 10: **Performance on general video understanding tasks.**

| Model | MVBench | VideoMME |
|---|---|---|
| VideoLLaMA2 | 54.6 | 46.6 |
| TRACE | 48.1 | 43.8 |
| TRACE-uni | 53.8 | 49.6 |

Table 11: **Performance on E.T.Bench.**

| E.T.Bench Metric | RAR Acc | ECA Acc | RVQ Acc | TVG F1 | EPM F1 | TAL F1 | EVS F1 | VHD F1 | DVC F1 | DVC Sim | SLC F1 | SLC Sim | TEM Rec | GVQ Acc |
|---|---|---|---|---|---|---|---|---|---|---|---|---|---|---|
| VideoLLama2 (7B) | 28.8 | 27.4 | 28.0 | 0.1 | 0.0 | 0.0 | 0.0 | 1.5 | 0.6 | 14.5 | 0.0 | 15.2 | 0.0 | - |
| Qwen2-VL (7B) | 39.4 | 34.8 | 42.2 | 3.9 | 0.1 | 0.3 | 0.4 | 20.6 | 0.0 | 0.0 | 0.0 | 0.0 | 6.6 | 55.9 |
| GPT-4o | 27.8 | 27.3 | 57.7 | 40.4 | 4.5 | 20.0 | 17.6 | 56.9 | 46.9 | 22.3 | 23.1 | 14.9 | 13.6 | - |
| TRACE (7B) | 29.4 | 28.8 | 42.6 | 46.8 | 12.3 | 21.6 | 26.6 | 45.2 | 45.7 | 24.0 | 27.3 | 17.7 | 17.8 | 52.4 |

**Effectiveness of TRACE on general video understanding tasks.** We evaluate TRACE on general video understanding tasks, and the results in Tables 10, 11, and 12 show that the TRACE architecture is still capable of handling general video understanding tasks and excel in VTG tasks:

- Despite not being trained on extensive multi-task datasets, TRACE is still highly effective in handling general video understanding tasks. For example, the TRACE outperform generalist video LLMs like VideoChat2, ShareGPT4Video, and ST-LLM on VideoMME benchmark.

- On the E.T.Bench (Liu et al.), TRACE outperforms VideoLlama2 across all tasks; achieves performance comparable to GPT-4o on RAR, ECA, RVQ, and DVC tasks; achieves similar performance to Qwen2-VL on RVQ and GVQ tasks; and outperforms both GPT-4o and Qwen2-VL on TVG, EPM, TAL, EVS, SLC, and TEM tasks.

- We train TRACE-uni by incorporating additional general video understanding data from a subset of LLaVA-Video-178k (Zhang et al., 2024)(specifically the perceptiontest and YouTube parts). TRACE-uni shows both improved general video understanding and stronger VTG performance without additional VTG training data.

- Notably, TRACE-uni performs on par with, or even outperforms, general video LLMs that use the same LLM backbone and vision encoder (VideoLlama2) using only about 2M training data. Additionally, TRACE-uni surpasses TRACE in VTG performance across all three evaluation datasets.

**Performance of TRACE with different number of slots per frame.** In Tables 13 and 14, we adopted the same settings as in Table 3, using VTG-IT only in Stage 2 and sampling 64 frames. Our findings are as follows: Increasing the number of slots per token significantly enhances TRACE's performance. Therefore, if computational or efficiency constraints are not a concern, we recommend using a larger number of slots per frame.

### B.3 CASE STUDIES

We present the case studies of TRACE in Figure 11. The results demonstrate that TRACE can accurately identify the events within the given video and is also proficient in performing traditional video captioning tasks.

Table 12: **Zero-shot performance of algorithms over various tasks.** We evaluated the performance of TRACE using the Youcook2, Charades-STA, and QVHighlights datasets. We highlight the best results for each block using **bold font**. The Valley, VideoChat-Embed, and Video-LLaMA results are elaborated from previous studies (Ren et al., 2023; Huang et al., 2023; Qian et al., 2024). The results with transparent text indicates unfair comparison (13B). We train TRACE-uni by incorporating additional general video understanding data from a subset of LLaVA-Video-178k (Zhang et al., 2024).

| Model | Youcook2 | | | Charades-STA | | QVHighlights | |
|---|---|---|---|---|---|---|---|
| | SODA_c | CIDEr | F1 Score | R@1$_{(IOU=0.5)}$ | R@1$_{(IOU=0.7)}$ | mAP | HIT@1 |
| *Traditional Video LLMs* | | | | | | | |
| Valley (7B) | 0.1 | 0.0 | 1.5 | 4.7 | 1.6 | 10.9 | 15.2 |
| VideoChat (7B) | 0.2 | 0.6 | 3.4 | 3.2 | 1.4 | 13.1 | 18.1 |
| Video-LLaMA (7B) | 0.0 | 0.0 | 0.1 | 2.7 | 1.2 | 11.3 | 15.6 |
| *Temporal Grounding Video LLMs* | | | | | | | |
| TimeChat (7B) | 1.2 | 3.4 | 12.6 | 32.2 | 13.4 | 14.5 | 23.9 |
| VTimeLLM (7B) | | | | 27.5 | 11.4 | | |
| VTimeLLM (13B) | | | | 34.3 | 14.7 | | |
| Momentor (7B) | | | | 26.6 | 11.6 | 7.6 | |
| HawkEye (7B) | | | | 31.4 | 14.5 | | |
| VTG-LLM (7B) | 1.5 | 5.0 | 17.5 | 33.8 | 15.7 | 16.5 | 33.5 |
| TRACE (7B) | 2.2 | 8.1 | **22.4** | 40.3 | 19.4 | 26.8 | 42.7 |
| TRACE-uni (7B) | **2.3** | **8.6** | **22.4** | **43.7** | **21.0** | **27.5** | **43.9** |

Table 13: **Performance of TRACE with different number of slots for each token on Youcook2 dataset.** We only use VTG-IT for training in Stage 2, and sample 64 frames for each video.

| Frame Num | Slot Num per Frame | SODA_c | CIDEr | F1 Score |
|---|---|---|---|---|
| 64 | 8 | 1.9 | 6.9 | 21.4 |
| 64 | 16 | 2.1 | 7.3 | 22.1 |

Table 14: **Performance of TRACE with different number of slots for each token on Youcook2 dataset.** We only use VTG-IT for training in Stage 2, and sample 64 frames for each video.

| Frame Num | Slot Num per Frame | R@1$_{IOU=0.5}$ | R@1$_{IOU=0.7}$ |
|---|---|---|---|
| 64 | 8 | 37.0 | 17.0 |
| 64 | 16 | 41.9 | 20.1 |

## C DISCUSSION ON CAUSAL EVENT MODELING

In this section, we provide a deeper analysis of the causal modeling approach presented in this paper. We also offer a comprehensive discussion and comparison of the following three key concepts:

- **Causal Language Modeling.** This is a well-established approach for decoder-only large language models (LLMs) and have become basis of current video LLMs (Li et al., 2023b; Ren et al., 2023; Huang et al., 2023)

- **Causal Event Modeling**. In contrast, TRACE introduces causal event modeling, which structures the responses of video LLMs by event triplets, providing a novel framework for understanding video content.

- **Complete Causal Relationship Modeling/Discovery.** Traditional video understanding models (Liang et al., 2022; Parmar et al., 2024; Chen et al., 2024c; Du et al., 2024; Yi et al., 2019; Girdhar & Ramanan, 2019; Li et al., 2020; Jin et al., 2022) focus extensively on discovering and analyzing the complex relationships between events within videos, offering a more comprehensive understanding of video content.

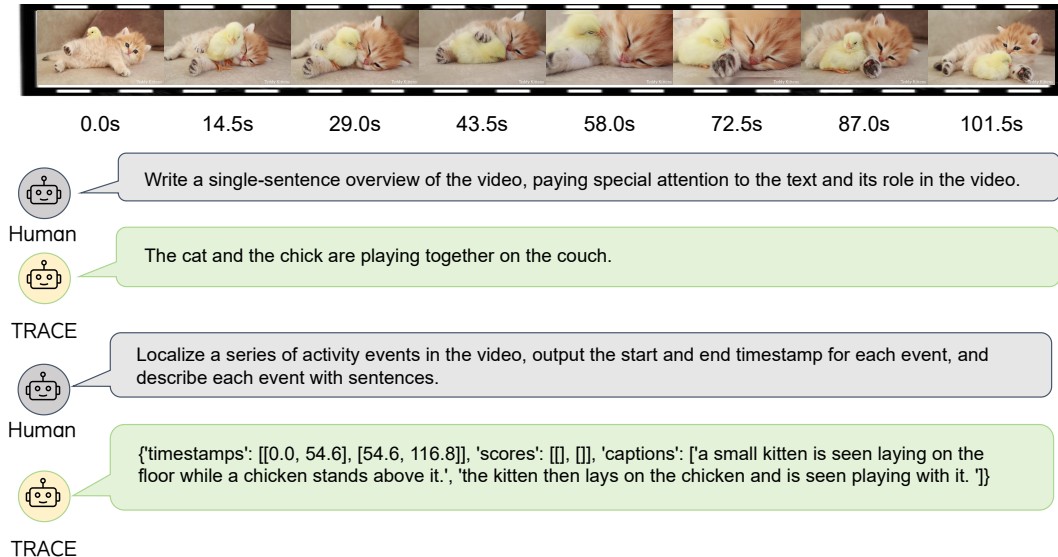

Figure 11: **Case study of TRACE.**

We would like to first discuss how causal event modeling excels in video understanding compared to causal language modeling. Next, we will explore the benefits, limitations, and potential integration of TRACE compared to complete causal relationship modeling/discovery methods.

## C.1 CAUSAL LANGUAGE MODELING VS. CAUSAL EVENT MODELING.

We will first provide a clear comparison between causal language modeling and causal event modeling, highlighting their similarities and differences. Then, we will delve into the benefits of using causal event modeling in more detail.

**Similarities.** Both causal language modeling and causal event modeling rely on decoder-only LLMs, which use video visual content ($F$) and instructions ($I$) as inputs. The models then generate responses based on the observations from both $I$ and $F$.

**Differences.** The key differences come from the output formats.

For causal language modeling, the response can be seen as a whole text content, i.e.,

$$p(\boldsymbol{Y}|\boldsymbol{F}, \boldsymbol{I}) = \prod_{y} p(y|y_i < y, \boldsymbol{F}, \boldsymbol{I}). \tag{3}$$

Here, timestamps, scores, and text are all represented by text tokens, and ordered following the natrual language structure.

For causal event modeling, the responses are formatted to series of event triplets $e_i = (t_i, s_i, c_i)$, i.e.,

$$p(\boldsymbol{R}|\boldsymbol{F}, \boldsymbol{I}) = \prod_{e_i} p(e_i|e_{1:i}, \boldsymbol{F}, \boldsymbol{I}) = \prod_{e_i} p(t_i|e_{1:i}, \boldsymbol{F}, \boldsymbol{I}) p(s_i|t_i, e_{1:i}, \boldsymbol{F}, \boldsymbol{I}) p(c_i|s_i, t_i, e_{1:i}, \boldsymbol{F}, \boldsymbol{I}). \tag{4}$$

**Key Improvements.** We can find that the key improvement of causal event modeling over causal language modeling comes from the format of model responses.

- **Clear inter- and intra- event triplets correlation.** The inter-event relationships are modeled through the next event prediction formulation. For intra-event relationships, we observe that a causal connection exists between $t_i \rightarrow s_i$, and between $(t_i, s_i) \rightarrow c_i$. In this context, $t_i$ and $s_i$ can also be viewed as a kind of thought chain when generating $c_i$.

- **Enable independent timestamps, scores, and text modeling.** We can separate the modeling of timestamps $t_i$, scores $s_i$, and text $c_i$, and independently model each component using different encoders and decoders. As directly adding new tokens to the text tokenizer may significantly disrupting the pretrained LLMs (Guo et al., 2024), such a decomposition helps to eliminate this issue, and the TRACE architecture retain the capacity on handling general video understanding tasks, as evaluated in Table 10.

## C.2 Causal Event Modeling vs. Complete Causal Relationship Modeling/Discovey

In this section, we will discuss the difference between causal event modeling and studies (Liang et al., 2022; Parmar et al., 2024; Chen et al., 2024c; Du et al., 2024; Yi et al., 2019; Girdhar & Ramanan, 2019; Li et al., 2020; Jin et al., 2022) focused on discovering or modeling complete causal relationships. Additionally, we will explore potential future work on integrating these two approaches.

**Discussion on studies that discovering or modeling complete causal relationships.** Based on the existing studies in this area, we classify the research into three main groups, which we will discuss in detail:

- **Building complete causal relationships to solve the video understanding problems.** VAR (Liang et al., 2022) uses an encoder-decoder architecture that first encodes video frames into event representations, then decodes each event into text captions. Similarly, Jin et al. (2022) constructs a learnable Markov Logic Network for action reasoning. Both of these approaches model complex causal relationships between events before generating answers, thereby enhancing the reasoning capacity of the models.
- **Introducing Benchmark Datasets for Complex Causal Reasoning.** Several studies have introduced benchmark datasets to evaluate models' ability to perform causal reasoning (Parmar et al., 2024; Yi et al., 2019; Girdhar & Ramanan, 2019; Du et al., 2024).
- **Discovering Causality from Videos.** Some research aims to automatically build causality graphs to uncover the relationships between events, providing a more comprehensive understanding of video content (Chen et al., 2024c; Li et al., 2020).

**Evaluation results of TRACE on causality reasoning benchmarks.** In Table 15, we present the performance of TRACE on the Event-Bench (Du et al., 2024).The results show that, despite being trained on less causal reasoning data (NextQA) compared to other video LLMs (Li et al., 2023c), TRACE outperforms open-source video LLMsand achieves performance comparable to GPT-4o on event description, contextural reasoning, and episodic reasoning tasks.

**Potential future improvements to TRACE through integration with causality discovery models.** Current video LLMs typically generate answers directly from text prompts and visual frames without explicitly modeling the causal relationships between events. While TRACE improves upon this limitation by representing model responses as event triplets, it still relies on previously generated events to produce subsequent triplets, due to the inherent architecture of decoder-only LLM backbones. To address this issue, further improvements could be made by integrating causality discovery methods:

- **Utilizing the outputs of causality discovery models as inputs for video LLMs.** For instance, we can encode the causality graph produced by Chen et al. (2024c); Li et al. (2020) as part of the model's inputs. For TRACE, this involves modifying $p(\boldsymbol{R}|\boldsymbol{F}, \boldsymbol{I})$ to $p(\boldsymbol{R}|\boldsymbol{F}, \boldsymbol{I}, \boldsymbol{C})$, where $\boldsymbol{C}$ represents the generated causality graph. By incorporating the causality graph as an additional input, the model would have access to richer context, enabling it to generate more accurate responses.
- **Utilizing the outputs of causality discovery models to construct Chain-of-Thought examples.** We can also guide video LLMs to first generate causality graphs before answering questions, similar to the approach used by Jin et al. (2022). For TRACE, we can modify $p(\boldsymbol{R}|\boldsymbol{F}, \boldsymbol{I})$ to $p(\boldsymbol{C}, \boldsymbol{R}|\boldsymbol{F}, \boldsymbol{I}) = p(\boldsymbol{C}|\boldsymbol{F}, \boldsymbol{I})p(\boldsymbol{R}|\boldsymbol{C}, \boldsymbol{F}, \boldsymbol{I})$.
- **Utilizing the outputs of causality discovery models to modify the attention masks of visual inputs.** Currently, the attention masks for visual inputs are typically designed the same as text

Table 15: **Evaluation results on Event-Bench.**

| Models | Atomic | Composite | | | Overall | | | | Avg |
|---|---|---|---|---|---|---|---|---|---|
| | Event Description | Temporal Reasoning | Causal Reasoning | Avg | Counter Reasoning | Contextual Reasoning | Episodic Reasoning | Avg. | |
| Video-LLaVA (7B) | 12.82 | 5.50 | 0.00 | 2.75 | 6.17 | 2.78 | 7.20 | 5.05 | 5.87 |
| VideoChat2 (7B) | 33.76 | 37.75 | 47.75 | 42.75 | 16.74 | 15.70 | 14.67 | 15.62 | 29.41 |
| ST-LLM (7B) | 47.22 | 48.75 | 59.50 | 54.13 | 9.69 | 25.32 | 16.67 | 18.66 | 37.71 |
| VIM (7B) | 48.08 | 51.25 | 61.25 | 56.25 | 22.91 | 32.66 | 18.67 | 25.71 | 41.64 |
| Gemini-1.5-Pro | 48.50 | 47.50 | 41.75 | 44.63 | 52.86 | 32.15 | 38.67 | 39.37 | 43.24 |
| GPT-4o | 54.27 | 56.75 | 58.25 | 57.50 | 63.44 | 50.13 | 37.33 | 49.24 | 53.33 |
| TRACE (7B) | 55.56 | 49.25 | 54.50 | 51.88 | 52.42 | 46.08 | 43.00 | 46.64 | 50.46 |

tokens, i.e., using causal attention masks. However, by incorporating the causality graph, we can enhance the attention masks by masking out visual tokens that are not causally related to the current reasoning task, thereby improving model focus on relevant events.

