# OpenReview forum: "TRACE: Temporal Grounding Video LLM  via Causal Event Modeling"
_ICLR.cc/2025/Conference — ICLR 2025 Poster_

### Official Review · Reviewer_YJgy · 2024-10-16

**Soundness:** 3
**Presentation:** 3
**Contribution:** 3
**Rating:** 8
**Confidence:** 5

**Summary:**

This paper proposes a new Video Temporal Grounding (VTG) method that addresses the shortcomings of existing LLM in handling VTG tasks by modeling causal events.

**Strengths:**

The author first employs a causal modeling method in the grounding of VLLM, achieving causal probability modeling through the sequence of input tokens. This approach will provide inspiration for future work on video understanding tasks using VLLM.

**Weaknesses:**

1. **Autoregressive modeling.**

One of my major concerns is that the authors have only used the earlier events $e_{1:k-1}$ in their modeling of causal relationships between events through autoregression, without incorporating the equally known $e_{k+1:K}$. I believe this approach may be unreasonable since it is likely that the same events may occur earlier while the current event is different due to unrelated pretexts. However, this issue can be avoided by modeling different subsequent events simultaneously. Besides, most current video understanding researchers have modeled multiple events by utilizing all contextual events that occur before and after them [1-4]. This may require the authors to provide further explanation.

[1] Liang C, et al. Visual abductive reasoning[C]. CVPR 2022.

[2] Lam T E, et al. CausalChaos! Dataset for Comprehensive Causal Action Question Answering Over Longer Causal Chains Grounded in Dynamic Visual Scenes. NeurIPS 2024.

[3] Chen T, et al. MECD: Unlocking Multi-Event Causal Discovery in Video Reasoning. NeurIPS 2024.

[4] Du Y, et al. Towards Event-oriented Long Video Understanding. ArXiv 2024.

2. **Inference speed.**

The authors have adopted a form similar to autoregression, and I would like to understand if there is a time overhead in comparing their model's inference speed to that of current mainstream LLMs.

3. **LLM backbone.**

I noticed that the authors used Mistral-7B as the LLM backbone, however, in other comparison methods, Timechat used LLaMA-2, while HawkEye, Momentor, and VTimeLLM used Vicuna. I would like to know if the authors have conducted experiments with LLaMA-2 or Vicuna as the LLM backbone, to ensure that the superior performance is not due to the better LLM backbone but rather the causal modeling.

**Questions:**

My main concern is with the autoregressive modeling approach, and if the authors can provide a reasonable explanation, I am willing to consider raising my score, as I believe this work could provide inspiration for future work.

---

> ### Author Response · Authors · 2024-11-17
> **Reply to Reviewer YJgy**
>
> Thank you for your valuable suggestions. We have addressed the questions you raised below.
> > **Autoregressive modeling.**
> >
>
> Thank you for the insightful suggestion! **After carefully examining the submission and the papers you listed, we acknowledge that the concept of "event" in our paper may differ from that in [1, 3], which could lead to some confusion.** To address this, we have revised the illustration and Eq. 1 of our paper to provide a clearer understanding of our approach.
>
> - **In summary, conditioned on prompts/instructions and all the video frame tokens, TRACE formats its responses using event triplets that consist of timestamps, scores, and text.** This structured approach aligns well with the inherent structure of videos and provides a comprehensive understanding of their content.
>     - In our work, we define an "event" as a triplet comprising timestamps, scores, and text. This triplet serves as a model output unit, providing textual descriptions (answers) along with the corresponding timestamps and matching scores.
>         - For dense video captioning tasks, the event triplets serve as descriptions that summarize the video contents (i.e., "events" in [1,3]).
>         - For moment retrieval tasks, the event triplets provide descriptions of specific moments along with their corresponding timestamps.
>         - For general video QA tasks, the event triplets may encompass the answer, accompanied by the relevant timestamps and matching scores. We believe that constructing these data is an important future work direction for TRACE.
>     - Importantly, each event $e_{k}$ is dependent on previous events $e_{1:k-1}$ and all the video visual contents $F$ (see Eq. 2). **By observing all the video visual inputs $F$ and previous generated events $e_{1:k-1}$, the model has sufficient information to generate the event $e_{k}$.**
>     - **In [1, 3], the events are considered as the fundamental units of video content, i.e., video clips.** The authors primarily focus on identifying the causality between these video clips. We believe that this approach is relatively orthogonal to ours. Moreover, the outputs generated by causality discovery models have the potential to be integrated as inputs of TRACE, which may further enhance its performance.
> - **The design of "casual event modeling" makes it easy to leverage pretrained LLMs.** Since current LLMs are typically decoder-only models, "casual event modeling" aligns closely with their pretraining objectives, allowing it to benefit from (1) instruction-following capabilities for handling diverse tasks simultaneously in a zero-shot manner, and (2) performance-boosting techniques like KV-cache and flash-attention. *While "casual event modeling" could be adapted to "masked event modeling" by taking into account future events, doing so would likely disrupt the pretrained knowledge of LLMs and result in a loss of the benefits.*
>
> > **Inference speed.**
> >
>
> Thank you for the suggestion! We have provided the inference speed of TRACE (128 frames) compared to VTG-LLM (96 frames), which is a typical video LLM with a standard architecture. The results demonstrate that TRACE does not incur any additional inference cost.
>
> | Youcook2 | Frame Num | Evaluation Time  | Throughout (seconds per token) | Memory (M) |
> | --- | --- | --- | --- | --- |
> | VTG-LLM | 96 | 2:01:28 | 0.0728 | 19063 |
> | TRACE | 128 | 1:40:40 | 0.0446 | 24007 |
>
> > **LLM backbone.**
> >
>
> Thank you for the valuable comments. Due to time and resource constraints, we have not conducted experiments using the llama2. However, we have conducted experiments without causal event modeling, also using Mixtral-7B as the LLM backbone.
>
> As shown in the "w/o causal event modeling" section of Table 4, following the previous SOTA [5], we (1) use slot-based compression, (2) add time tokens and score tokens to the text vocabulary, and (3) use natural language outputs instead of event-structured outputs. The training data, vision encoder, and LLMs remain the same as in the original TRACE. **The results show that TRACE significantly outperforms this ablation, even with fewer sampled frames.**
>
> [5] VTG-LLM: Integrating Timestamp Knowledge into Video LLMs for Enhanced Video Temporal Grounding. Arxiv 2024.

---

> > ### Comment · Reviewer_YJgy · 2024-11-17
> > **Official Comment by Reviewer YJgy**
> >
> > Thanks for the further claim, part of my concerns has been addressed. However, given previous work on combining causal modeling with video understanding, I still believe that modeling only earlier events represents a suboptimal modeling approach. **Despite recognizing that this limitation may be attributed to the inherent properties of decoder-only LLMs and acknowledging the authors' innovative and insightful endeavors, I remain skeptical of the theoretical rigor.**
> >
> > According to [1-4], as well as [6-9], modeling the complete causal relationship between events is essential for comprehensive video content understanding. Moreover, as indicated in [6], the video's Predictive Questions Test is a crucial indicator of causality. Therefore, as Reviewer p7QK, MMw2 points out, I believe that the modeling method of TRACE may not perform well in tasks such as VQA, particularly in predictive questions.  This limitation reduces its inspiration to other video understanding tasks, which I initially recognized and appreciated about this work (contribution to video understanding + causal reasoning).
> >
> > Moreover, I believe there is a significant correlation between VTG, VQA, and DVC, rather than orthogonality. I remain firm in my belief that the approach to causal modeling needs to be discussed with current video understanding + causal reasoning works in [1-4] and [6-9], either in the related work or other sections.
> >
> > Therefore, I maintain my current rating.
> >
> > [6] Clevrer: Collision events for video representation and reasoning. ICLR 2020.
> >
> > [7] CATER: A diagnostic dataset for Compositional Actions and Temporal Reasoning. ICLR 2020.
> >
> > [8] Causal discovery in physical systems from videos. NeurIPS 2020.
> >
> > [9] Complex Video Action Reasoning via Learnable Markov Logic Network. CVPR 2022.

---

> ### Author Response · Authors · 2024-11-17
> **Further Reply to Reviewer YJgy**
>
> Thank you for your prompt response and insightful suggestions. We fully acknowledge the importance of modeling the complete causal relationship between events for a comprehensive understanding of video content, as you have pointed out. We would like to provide further clarification regarding our study.
> - Firstly, while **modeling the complete relationship may pose a challenge for decoder-only LLMs, this is a general limitation, not specific to TRACE.** Given that most current video LLMs also rely on decoder-only architectures, we believe that TRACE's structure does not diminish its capacity for general VQA tasks compared to these approaches. We have previously reported this to reviewers p7QK and MMw2 during the rebuttal phase.
> - Secondly, we agree with your observation that causality discovery is not orthogonal to VTG tasks. We believe that it is possible to design causality discovery models in an 'orthogonal' manner, which can provide a comprehensive understanding of video content. The outputs of these causality discovery models can then be used as supplementary inputs for TRACE, potentially overcoming some of the limitations associated with decoder-only LLMs. We have discussed these related works and future directions in the 'Conclusion and Future Works' section of revised paper.
> - Lastly, despite TRACE's potential limitations in modeling the complete causal relationship, we firmly believe that it still provides significant benefits. By representing the response of video LLMs as a series of event triplets (timestamps, scores, and captions), TRACE excels in VTG tasks. Furthermore, we envision that TRACE can be extended to other video understanding tasks, such as VQA, by expanding its annotations in the future. Therefore, we believe that TRACE remains a valuable contribution to the field of video understanding.
>
> Thank you once again for your detailed review, which has greatly enhanced the clarity of our paper.

---

> > ### Comment · Reviewer_YJgy · 2024-11-17
> >
> > Thanks to the author for further reply. In my opinion, it is necessary to conduct a comparison and further analysis between VideoLLM-based causal modeling and traditional video understanding model, which will help readers better understand this autoregressive causal modeling and make the causal method proposed in this paper more rigorous in theory. I look forward to the further discussion of video causal reasoning work [1-4, 6-9] etc. by the author in the revised draft. As such, I think this article will provide insights not only for simple VTG task, but also for the field of video understanding, and I am willing to raise my score to 8 if the further analysis is conducted soundly.

---

> > > ### Author Response · Authors · 2024-11-18
> > > **Revision for Detailed Discussion.**
> > >
> > > Dear Reviewer YJgy,
> > >
> > > Thank you very much for your thoughtful follow-up suggestions! We have revised our paper to include a detailed discussion comparing (1) causal language modeling (e.g., Video LLMs), (2) causal event modeling (TRACE), and (3) causality modeling/discovery models ([1-4, 6-9]). Below are the key takeaways, and we encourage you to refer to Appendix C for the full discussion:
> > > - TRACE enhances causal language modeling by:
> > >     - Providing clear correlations inter- and intra-event triplets.
> > >     - Independently modeling timestamps, scores, and textual information.
> > > - We have also expanded the discussion on related works, including:
> > >     - Building complete causal relationships to address video understanding problems ([1, 9]).
> > >     - Introducing benchmark datasets for complex causal reasoning ([2, 4, 6, 7]).
> > >     - Discovering causality in videos ([3, 8]).
> > > - We evaluated TRACE on causality reasoning benchmark [4], where it outperformed open-source video LLMs and achieved performance comparable to GPT-4o on tasks such as event description, contextual reasoning, and episodic reasoning. Although we also attempted to evaluate TRACE on [2], the raw video data was not provided.
> > > - In addition, we discussed potential future improvements to TRACE through integration with causality discovery models, including:
> > >     - Using outputs from causality discovery models as inputs for video LLMs.
> > >     - Leveraging causality discovery outputs to construct Chain-of-Thought examples.
> > >     - Applying causality discovery outputs to modify attention masks for visual inputs.
> > >
> > > We hope the revised discussions and results address your concerns. Please feel free to reach out if you have any additional questions. Thank you again for your time and valuable feedback!

---

> > > > ### Comment · Reviewer_YJgy · 2024-11-19
> > > > **Further Reply by Reviewer YJgy**
> > > >
> > > > Thanks for the further reply.
> > > >
> > > > A minor mistake: On lines 1253-1254, I noticed that the training data for TRACE includes Next-QA (a temporal reasoning dataset). Hence, the statement "despite not being trained on large-scale causal reasoning datasets" needs to be modified.
> > > > However, compared to VideoChat2, TRACE uses fewer training data (e.g. CLEVRER), so the experimental results still prove the effectiveness of TRACE's causal modeling method.
> > > >
> > > > My main concerns have been addressed, and I hope the authors ensure that this analysis section will be included in the final version. I believe that TRACE will bring insights to the field of video understanding and provide inspiration for causal modeling in VideoLLMs.
> > > >
> > > > In conclusion, I recommend accepting this paper.

---

> > > > > ### Author Response · Authors · 2024-11-19
> > > > > **To Reviewer YJgy**
> > > > >
> > > > > Dear Reviewer YJgy,
> > > > >
> > > > > Thank you for raising your score! We will incorporate the discussion section in the final version of the paper. Additionally, we are continuing to refine the main sections to include key takeaways of the discussion. Your detailed review has been invaluable in helping us improve the quality of the paper.

---

### Official Review · Reviewer_MMw2 · 2024-10-20

**Soundness:** 3
**Presentation:** 3
**Contribution:** 3
**Rating:** 5
**Confidence:** 4

**Summary:**

This paper addresses the task of Video Temporal Grounding (VTG) and introduces TRACE, a task-interleaved Video-LLM designed for enhanced VTG performance. The authors highlight limitations in current Video-LLMs, which rely solely on natural language generation without considering the inherent temporal structure of videos. To address this, they propose a novel causal event modeling framework, decomposing videos into sequences of events defined by timestamps, salient scores, and textual captions. Extensive experiments on datasets such as Charades-STA, QVHighlights, and YouCook2 demonstrate the superior zero-shot performance of TRACE compared to existing Video-LLMs.

**Strengths:**

1) Video Temporal Grounding (VTG) is a crucial task, yet current Video-LLMs underperform in this area. Techniques aimed at improving temporal grounding for these models are highly valuable to advance the field.
2) The causal event modeling framework fits well with the next-token prediction paradigm of large language models (LLMs), offering an intuitive way to model video structures in sequential tasks.
3) TRACE demonstrates consistent performance improvements over prior Video-LLMs across three key VTG benchmarks (Charades-STA, QVHighlights, and YouCook2), underscoring its effectiveness.

**Weaknesses:**

1) While the motivation for TRACE is clear, the use of multiple task-specific heads may limit the model’s generalization. A primary appeal of Video-LLMs lies in their ability to handle a variety of tasks without specific fine-tuning. TRACE’s focus on VTG may narrow its versatility, making it less effective for general video understanding tasks. In most cases, lightweight VTG-specific models with stronger performance could be more suitable for VTG scenarios.
2) Some clarity is not clear. For example, the paper does not adequately explain slot-based compression, which is not a widely known technique. Moreover, compressing each frame to just 8 visual tokens might lead to significant information loss, raising concerns about the trade-off between efficiency and accuracy.
3) It is unclear whether the same set of number tokens is used for both timestamps and scores. If so, this could blend the two types of information, contradicting the authors' claim (lines 45–46) that the model preserves the distinct structure of video events.

**Questions:**

See Weaknesses

---

> ### Author Response · Authors · 2024-11-17
> **Reply to Reviewer MMw2 (1/2)**
>
> Thank you for your valuable suggestions. We have addressed the questions you raised below.
>
> > While the motivation for TRACE is clear, the use of multiple task-specific heads may limit the model’s generalization. A primary appeal of Video-LLMs lies in their ability to handle a variety of tasks without specific fine-tuning. TRACE’s focus on VTG may narrow its versatility, making it less effective for general video understanding tasks.
> >
>
> Thank you for the insightful suggestion! We have also recognized this limitation and conducted additional experiments before the rebuttal. **The results show that the TRACE architecture is still capable of handling general video understanding tasks and excel in VTG tasks.**
>
> - *Despite **NOT** being trained on extensive multi-task datasets,* TRACE is still highly effective in handling general video understanding tasks. For example, the TRACE outperform generalist video LLMs like VideoChat2, ShareGPT4Video, and ST-LLM on VideoMME benchmark.
> - We train TRACE-uni by incorporating additional general video understanding data from a subset of LLaVA-Video-178k (specifically the perceptiontest and YouTube parts). **TRACE-uni shows both improved general video understanding and stronger VTG performance without additional VTG training data.** Notably,
>     - TRACE-uni performs on par with, or even outperforms, generalist video LLMs that use the same LLM backbone and vision encoder (VideoLlama2) using only about 2M training data.
>     - TRACE-uni surpasses TRACE in VTG performance across all three evaluation datasets.
>
> | MVBench | Avg | AS | AP | AA | FA | UA | OE | OI | OS | MD | AL | ST | AC | MC | MA | SC | FP | CO | EN | ER | CI |
> | --- | --- | --- | --- | --- | --- | --- | --- | --- | --- | --- | --- | --- | --- | --- | --- | --- | --- | --- | --- | --- | --- |
> | VideoLLama2 | 54.6 |  |  |  |  |  |  |  |  |  |  |  |  |  |  |  |  |  |  |  |  |
> | TRACE | 48.1 | 61.2 | 56.5 | 72.5 | 46.5 | 61.0 | 48.0 | 69.5 | 40.0 | 22.0 | 31.0 | 86.5 | 37.5 | 37.0 | 51.0 | 45.0 | 40.5 | 39.0 | 31.0 | 43.5 | 44.5 |
> | TRACE-uni | 53.8 | 68.1 | 58.5 | 72.5 | 41.5 | 73.5 | 55.1 | 71.5 | 40.5 | 25.0 | 53.0 | 88.5 | 63.5 | 38.5 | 51.0 | 52.5 | 49.0 | 59.5 | 33.5 | 49.5 | 32.5 |
>
> | VideoMME (w/o subtitle) | Short | Midium | Long | Avg |
> | --- | --- | --- | --- | --- |
> | VideoLLama2 |  |  |  | 46.6 |
> | TRACE | 49.5 | 42.5 | 39.3 | 43.8 |
> | TRACE-uni | 58.2 | 48.1 | 42.3 | 49.6 |
>
> | Youcook2 (Zero-Shot) | CIDER | METEOR | SODA_c | F1 |
> | --- | --- | --- | --- | --- |
> | TRACE | 8.1 | 2.8 | 2.2 | 22.4 |
> | TRACE-uni | 8.6 | 2.9 | 2.3 | 22.4 |
>
> | Charades-STA (Zero-Shot) | 0.3 | 0.5 | 0.7 | mIOU |
> | --- | --- | --- | --- | --- |
> | TRACE | 58.6 | 40.3 | 19.4 | 38.7 |
> | TRACE-uni | 63.7 | 43.7 | 21.0 | 41.5 |
>
> | QVHighlights (Zero-Shot) | mAP | Hit@1 |
> | --- | --- | --- |
> | TRACE | 26.8 | 42.7 |
> | TRACE-uni | 27.5 | 43.9 |
>
> ---
>
> > In most cases, lightweight VTG-specific models with stronger performance could be more suitable for VTG scenarios.
> >
>
> Thank you for your insightful comments. We would like to clarify a few points and provide further context:
>
> - Video LLMs offer distinct advantages that traditional VTG-specific models cannot match. Specifically, they provide:
>     1. *Zero-shot capability*, which allows the models to perform VTG tasks without the need for task-specific training.
>     2. *The ability to handle multiple VTG tasks simultaneously*, offering a level of versatility that traditional models lack.
>
>     *We believe these features are crucial for real-world applications, where scalability and adaptability are keys.*
>
> - While VTG-specific models generally exhibit strong performance on VTG tasks, there is an emerging research trend exploring how video LLMs can address VTG challenges [1, 2, 3, 4, 5]. *TRACE introduces a novel solution that significantly enhances the performance of video LLMs on VTG tasks, and we believe this contribution is valuable to the community.*

---

> ### Author Response · Authors · 2024-11-17
> **Reply to Reviewer MMw2 (2/2)**
>
> > Some clarity is not clear. For example, the paper does not adequately explain slot-based compression, which is not a widely known technique. Moreover, compressing each frame to just 8 visual tokens might lead to significant information loss, raising concerns about the trade-off between efficiency and accuracy.
> >
>
> Thank you for your valuable suggestion! We would like to provide the following clarification
>
> - *We compress the visual tokens to address efficiency and context length limitations.* Since TRACE samples 128 frames, without compression, the ViT would produce over 70K visual tokens. To handle this, we compress the visual tokens to 8 tokens per frame, resulting in a total of 1,792 visual tokens after incorporating the time tokens corresponding to each frame. This compression allows us to effectively handle VTG tasks within the 4K context length limit.
> - *We choose slot-based compression for its lightweight architecture and high performance on VTG tasks.* Introduced by [5], slot-based compression uses only one-third of the parameters of a single cross-attention layer, while outperforming both cross-attention and sampling-based methods on VTG tasks.
> - As per your recommendation, we have conducted ablation studies on the number of tokens per frame. **However, the training will take more than a week to complete. We will post the results here once the ablation training is finished.**
>
> > It is unclear whether the same set of number tokens is used for both timestamps and scores. If so, this could blend the two types of information, contradicting the authors' claim (lines 45–46) that the model preserves the distinct structure of video events.
> >
>
> We apologize for any confusion. **To clarify, timestamps and scores in TRACE use separate sets of tokens.** As explained in lines 235-245, the timestamps and scores are processed by distinct encoder-decoder pairs, though both pairs share the same model structural design.
>
> [1] Vtimellm: Empower llm to grasp video moments. CVPR 2024.
>
> [2] Timechat: A time-sensitive multimodal large language model for long video understanding. CVPR 2024.
>
> [3] Lita: Language instructed temporal-localization assistant. ECCV 2024.
>
> [4] Momentor: Advancing Video Large Language Model with Fine-Grained Temporal Reasoning. ICML 2024.
>
> [5] VTG-LLM: Integrating Timestamp Knowledge into Video LLMs for Enhanced Video Temporal Grounding. Arxiv 2024.

---

> ### Author Response · Authors · 2024-11-19
> **Results on number of slots per frame**
>
> Dear Reviewer MMw2,
>
> We have finished the experiments that compressing each frame into 16 tokens. Due to time constraints, we adopted the same settings as in Table 3, using VTG-IT only in Stage 2 and sampling 64 frames. Our findings are as follows: Increasing the number of slots per token significantly enhances TRACE's performance. Therefore, if computational or efficiency constraints are not a concern, we recommend using a larger number of slots per frame.
>
> | Youcook2 | Frame Num | Slot Num per Frame | SODA_c | CIDEr |  F1 Score |
> | --- | --- | --- | --- | --- | --- |
> |  | 64 | 8 | 1.9 | 6.9 | 21.4 |
> |  | 64 | 16 | 2.1 | 7.3 | 22.1 |
>
> | Charades-STA | Frame Num | Slot Num per Frame | R@1$_{IOU=0.5}$ | R@1$_{IOU=0.7}$ |
> | --- | --- | --- | --- | --- |
> |  | 64 | 8 | 37.0 | 17.0 |
> |  | 64 | 16 | 41.9 | 20.1 |
>
> Additionally, we evaluated TRACE on the causality reasoning benchmark [6], as shown in Table 11 of the revised paper. TRACE outperformed open-source video LLMs and achieved performance comparable to GPT-4o on tasks such as event description, contextual reasoning, and episodic reasoning. This result demonstrates the potential of the TRACE architecture for handling complex reasoning tasks.
>
> We hope our responses have sufficiently addressed your concerns. Please feel free to reach out if you have any further questions. Thank you again for your time and effort in reviewing our paper.
>
> [6] Towards Event-oriented Long Video Understanding. ArXiv 2024.

---

> > ### Comment · Reviewer_MMw2 · 2024-11-22
> >
> > I appreciate the author’s response and their efforts to provide additional comparisons and clarifications. While some of the points raised in my original review were partially addressed, there remain core concerns that were not fully resolved. These issues are also echoed by other reviewers (e.g., KYwb).
> >
> > One key concern is that the main argument regarding causal event modeling is still weakly supported. It remains unclear why the authors chose to focus on causal event modeling as the primary approach for structuring video representations. Videos inherently comprise diverse components—such as objects, backgrounds, actions, and interactions—that extend beyond salient scores and timestamps. While I understand the authors’ intention to draw inspiration from causal language modeling, the analogy appears to lack a solid foundation. Unlike language, which is relatively homogeneous and well-suited to the next-token prediction paradigm, the relationship between language, salient scores, and timestamps is less evident. Additionally, the necessity of a dedicated time/score head is questionable. Why not directly integrate these aspects into the text token space for modeling?
> >
> > The ablation studies on the slot-based compression are appreciated. However, the additional results mainly show that increasing the number of tokens improves performance. This neither demonstrates the advantages of the proposed approach over established techniques such as Q-Former or 2D Average Pooling nor suggests a number of 8/16 tokens per frame is enough for video modeling. While the results on MVBench and VideoMME are promising, they remain significantly behind the performance of popular models like LLaVA-Onevision or Qwen2-VL.
> >
> > Although I will maintain my score currently, I strongly encourage the authors to incorporate the suggested improvements into their revision. I thank the authors for their efforts and look forward to further refinements that could address these fundamental concerns.

---

> > > ### Author Response · Authors · 2024-11-23
> > > **Further Clarification to Reviewer MMw2 (1/2)**
> > >
> > > We thank the reviewer for their further response. However, we would like to provide additional clarifications and will incorporate the following discussions and explanations in the revised paper.
> > >
> > > > One key concern is that the main argument regarding causal event modeling is still weakly supported. It remains unclear why the authors chose to focus on causal event modeling as the primary approach for structuring video representations. Videos inherently comprise diverse components—such as objects, backgrounds, actions, and interactions—that extend beyond salient scores and timestamps. While I understand the authors’ intention to draw inspiration from causal language modeling, the analogy appears to lack a solid foundation. Unlike language, which is relatively homogeneous and well-suited to the next-token prediction paradigm, the relationship between language, salient scores, and timestamps is less evident.
> > >
> > > Thank you for the insightful discussion, and we agree with the reviewer that videos contain many components. However, we would like to clarify the following points, and will incorporate the discussion in the revised paper:
> > >
> > > - **We believe that the event triplet (timestamp, score, language) naturally arises in video LLM responses**, as every video content, including the objects, actions, and interactions mentioned by the reviewers, inherently has a temporal component (i.e., a happening time). The input query and the corresponding video content will also have associated scores.
> > > - **We would like to clarify that the primary focus of this paper is on VTG tasks**, which is why we concentrate mainly on temporal aspects, specifically timestamps and salient scores. For other aspects, such as objects, actions, and interactions, we continue to rely on language modeling. *While future work could explore separate or causal modeling for these components, as well as extending the framework to areas like object detection and interaction recognition, these topics may beyond the scope of our current study.*
> > > - We have provided additional discussion on the benefits of our framework in lines 1229-1241 of the revised paper. In summary, the key improvement of causal event modeling over causal language modeling lies in the structure of the model's responses: (1) Explicit cross-triplet correlations; (2) Enabling independent modeling of timestamps, scores, and text.
> > >
> > > > The necessity of a dedicated time/score head
> > > >
> > >
> > > Thank you for your comments. We would like to clarify the following points:
> > >
> > > - **Using separate time tokens can improve the performance of VTG tasks**, a finding that has been widely verified in previous studies [3, 4, 5, 7]. These studies demonstrate that explicitly modeling temporal information with dedicated tokens enhances the model's ability to handle time-dependent tasks effectively.
> > > - **As pointed out by the reviewer, language is relatively homogeneous and well-suited to next-token prediction, whereas timestamps and scores may not align well with the language space.** Previous studies [5] have shown that adding time tokens to the language model can hurt language generation capabilities. To address this issue, we model language, timestamps, and scores in separate spaces, and decoding them one by one, thereby avoiding this weakness.
> > > - **We have conducted ablation experiments**
> > >     - Using causal language modeling with only the text tokenizer/head.
> > >     - Using causal event modeling without separate, dedicated time/score heads.
> > >
> > >     As shown in Table 3 of our submission:
> > >
> > >     - Using causal language modeling significantly reduces model performance, resulting in a 2.6-point drop in the CIDEr metric on YouCook2 and a 7.3% performance drop in the R@1$_{IOU=0.5}$ metric for Charades-STA.
> > >     - When using causal event modeling with a shared head for timestamps, scores, and language, the pretrained knowledge of the LLM is disrupted. This disruption causes the model to fail in following instructions and prevents it from completing the evaluation task.
> > >
> > > [7] Vid2seq: Large-scale pretraining of a visual language model for dense video captioning. CVPR 2023.

---

> > > ### Author Response · Authors · 2024-11-23
> > > **Further Clarification to Reviewer MMw2 (2/2)**
> > >
> > > > However, the additional results mainly show that increasing the number of tokens improves performance. This neither demonstrates the advantages of the proposed approach over established techniques such as Q-Former or 2D Average Pooling nor suggests a number of 8/16 tokens per frame is enough for video modeling.
> > > >
> > >
> > > We are sorry for the misunderstanding, and would like to clarify the following points:
> > >
> > > - **We would like to clarify that the slot-based compression method was not proposed by this paper.** This method was introduced in [5], and we agree with the reviewer that alternative approaches may offer better performance. However, we believe that further verifying the effectiveness of slot-based compression over other methods is responsible for [5] instead of this paper.
> > > - We believe it is clear that increasing the number of tokens per frame leads to better model performance. **If computational resources and the context length permit, we recommend avoiding visual token compression.**
> > > - **The main contribution of this paper lies in the causal event modeling framework** **and the task-interleaved structure.** We believe this contribution is orthogonal to the design of compression layers.
> > >
> > > > While the results on MVBench and VideoMME are promising, they remain significantly behind the performance of popular models like LLaVA-Onevision or Qwen2-VL.
> > > >
> > >
> > > Thank you for the detailed comments. We believe it is **unfair** to expect TRACE to achieve comparable performance to popular models like LLaVA-Onevision or Qwen2-VL on general video understanding tasks for the following reasons:
> > >
> > > - **LLaVA-Onevision and Qwen2-VL leverage more advanced LLM backbones (such as Qwen2), larger training datasets (over 5 million samples), and longer context lengths (greater than 8K).** These factors significantly contribute to their stronger benchmark performance. In contrast, our model is trained with a 4K context length, uses the Mistral-7B-v0.2 LLM backbone, and relies on a dataset of just 2 million SFT examples, of which only 1 million are general video understanding samples (TRACE-uni). Given these differences, we believe the comparison is not entirely fair.
> > > - **TRACE-uni has achieved performance on par with, or even surpassing, VideoLLama2 (which uses the same LLM backbone and vision encoder)** on general video understanding tasks. Moreover, TRACE-uni was trained **with only about 1M general video understanding task training data**, a significantly smaller dataset compared to VideoLLama2.
> > >
> > > Thank you once again for your thoughtful reviews. We hope the discussions and clarifications provided above address your concerns.

---

> ### Author Response · Authors · 2024-11-26
> **Additional Evaluation Results**
>
> Dear reviewer MMw2,
>
> To further clarify our points, we evaluate Qwen2-VL on the **event-level video understanding benchmark** **E.T.Bench** [8]. Our results show that, *despite being trained with less data, shorter context, and an older LLM backbone*
>
> - On the **E.T. Bench**, TRACE
>     - outperforms VideoLlama2 across all tasks;
>     - achieves performance comparable to GPT-4o on RAR, ECA, RVQ, and DVC tasks;
>     - achieves similar performance to Qwen2-VL on RVQ and GVQ tasks;
>     - outperforms both GPT-4o and Qwen2-VL on TVG, EPM, TAL, EVS, SLC, and TEM tasks.
> - While **Qwen2-VL** demonstrates advanced performance on multiple-choice QA tasks such as the RAR, ECA, and RVQ evaluations from E.T.Bench, outperforming even GPT-4. But its performance on other tasks, ranging from TVG to TEM, remains subpar. This highlights the ongoing challenge of developing a generalist video LLM capable of effectively handling a wide variety of video understanding tasks.
>
> | E.T.Bench | RAR,Acc | ECA,Acc | RVQ,Acc | TVG,F1 | EPM,F1 | TAL,F1 | EVS,F1 | VHD,F1 | DVC,F1 | DVC,Sim | SLC,F1 | SLC,Sim | TEM,Rec | GVQ,Acc |
> | --- | --- | --- | --- | --- | --- | --- | --- | --- | --- | --- | --- | --- | --- | --- |
> | VideoLLama2 (7B) | 28.8 | 27.4 | 28.0 |  0.1 |  0.0 | 0.0 |  0.0 |  1.5 |  0.6 |  14.5 | 0.0 |  15.2 | 0.0 | - |
> | Qwen2-VL (7B) | 39.4 | 34.8 | 42.2 | 3.9  | 0.1  | 0.3  | 0.4  | 20.6 | 0.0 | 0.0 | 0.0 | 0.0 | 6.6 | 55.9 |
> | GPT-4o | 27.8 | 27.3 | 57.7 | 40.4 | 4.5 | 20.0 | 17.6 | 56.9 | 46.9 | 22.3 | 23.1 | 14.9 | 13.6 | - |
> | TRACE (7B) | 29.4  | 28.8  | 42.6  | 46.8  | 12.3  | 21.6  | 26.6  | 45.2 | 45.7  | 24.0  | 27.3  | 17.7  | 17.8  | 52.4 |
>
> We hope the additional numerical results address your concerns. Please don't hesitate to reach out if you have any further questions. Thank you again for your review.
>
> [8] E.T. Bench: Towards Open-Ended Event-Level Video-Language Understanding. NeurIPS 2024.

---

### Official Review · Reviewer_p7QK · 2024-10-27

**Soundness:** 3
**Presentation:** 2
**Contribution:** 2
**Rating:** 6
**Confidence:** 3

**Summary:**

The paper introduces a task-interleaved video LLM, TRACE, which incorporates a newly-designed causal event modeling framework for VTG task. The TRACE employs multiple encoders for different inputs, while the task tokens are arranged in an interleaved manner. TRACE demonstrates SOTA performance on various VTG datasets compared to previous video LLMs.

**Strengths:**

1. The presentation and illustration are quite clear and easy to follow.
2. The motivation of causal event modeling is quite intuitive and the design is straightforward and yet effective.
3. The zero-shot performance is superior compared to previous video LLM methods.

**Weaknesses:**

1. While the paper compares TRACE with other video LLMs, it presents limited comparison and may not adequately address how it stands against traditional non-generative and task-specific models.
2. The extent to which TRACE can be applied to other types of video tasks beyond VTG is unclear. Its design may be highly specialized, which could limit its applicability across diverse video understanding tasks. Authors should present more results on other video-understanding tasks since the design seems generalizable by building such causal event relations.

**Questions:**

See weaknesses.

---

> ### Author Response · Authors · 2024-11-17
> **Reply to Reviewer p7QK**
>
> Thank you for your valuable suggestions. We have addressed the questions you raised below.
>
> > While the paper compares TRACE with other video LLMs, it presents limited comparison and may not adequately address how it stands against traditional non-generative and task-specific models.
> >
>
> Thank you for the detailed comment. We have provided TRACE’s results compared to traditional non-generative and task-specific models after fine-tuning in Table 5. The results show that:
>
> - *TRACE sets a new SOTA on the YouCook2 dataset without audio inputs,* outperforming existing SOTA by a large margin, and even surpassing Vid2Seq (with audio) on F1 score.
> - *On the Charades-STA dataset, TRACE performs comparably to non-generative methods* and outperforms strong baselines such as VDI, Moment-DETR, and UnLoc-L.
> - *TRACE significantly outperforms other video LLMs after fine-tuning,* highlighting the potential of video LLMs for VTG tasks.
>
> Beyond fine-tuned results, **TRACE’s strong zero-shot performance, surpassing existing methods and handling multiple VTG tasks simultaneously, offers significant value**—something traditional non-generative and task-specific models cannot achieve.
>
> > The extent to which TRACE can be applied to other types of video tasks beyond VTG is unclear. Its design may be highly specialized, which could limit its applicability across diverse video understanding tasks. Authors should present more results on other video-understanding tasks since the design seems generalizable by building such causal event relations
> >
>
> Thank you for the insightful suggestion! We have also recognized this limitation and conducted additional experiments before the rebuttal. **The results show that the TRACE architecture is still capable of handling general video understanding tasks and excel in VTG tasks:**
>
> - *Despite **NOT** being trained on extensive multi-task datasets,* TRACE is still highly effective in handling general video understanding tasks.  For example, the TRACE outperform generalist video LLMs like VideoChat2, ShareGPT4Video, and ST-LLM on VideoMME benchmark.
> - We train TRACE-uni by incorporating additional general video understanding data from a subset of LLaVA-Video-178k (specifically the perceptiontest and YouTube parts). **TRACE-uni shows both improved general video understanding and stronger VTG performance without additional VTG training data.** Notably,
>     - TRACE-uni performs on par with, or even outperforms, general video LLMs that use the same LLM backbone and vision encoder (VideoLlama2) using only about 2M training data.
>     - TRACE-uni surpasses TRACE in VTG performance across all three evaluation datasets.
>
>
> | MVBench | Avg | AS | AP | AA | FA | UA | OE | OI | OS | MD | AL | ST | AC | MC | MA | SC | FP | CO | EN | ER | CI |
> | --- | --- | --- | --- | --- | --- | --- | --- | --- | --- | --- | --- | --- | --- | --- | --- | --- | --- | --- | --- | --- | --- |
> | VideoLLama2 | 54.6 |  |  |  |  |  |  |  |  |  |  |  |  |  |  |  |  |  |  |  |  |
> | TRACE | 48.1 | 61.2 | 56.5 | 72.5 | 46.5 | 61.0 | 48.0 | 69.5 | 40.0 | 22.0 | 31.0 | 86.5 | 37.5 | 37.0 | 51.0 | 45.0 | 40.5 | 39.0 | 31.0 | 43.5 | 44.5 |
> | TRACE-uni | 53.8 | 68.1 | 58.5 | 72.5 | 41.5 | 73.5 | 55.1 | 71.5 | 40.5 | 25.0 | 53.0 | 88.5 | 63.5 | 38.5 | 51.0 | 52.5 | 49.0 | 59.5 | 33.5 | 49.5 | 32.5 |
>
> | VideoMME (w/o subtitle) | Short | Midium | Long | Avg |
> | --- | --- | --- | --- | --- |
> | VideoLLama2 |  |  |  | 46.6 |
> | TRACE | 49.5 | 42.5 | 39.3 | 43.8 |
> | TRACE-uni | 58.2 | 48.1 | 42.3 | 49.6 |
>
> | Youcook2 (Zero-Shot) | CIDER | METEOR | SODA_c | F1 |
> | --- | --- | --- | --- | --- |
> | TRACE | 8.1 | 2.8 | 2.2 | 22.4 |
> | TRACE-uni | 8.6 | 2.9 | 2.3 | 22.4 |
>
> | Charades-STA (Zero-Shot) | 0.3 | 0.5 | 0.7 | mIOU |
> | --- | --- | --- | --- | --- |
> | TRACE | 58.6 | 40.3 | 19.4 | 38.7 |
> | TRACE-uni | 63.7 | 43.7 | 21.0 | 41.5 |
>
> | QVHighlights (Zero-Shot) | mAP | Hit@1 |
> | --- | --- | --- |
> | TRACE | 26.8 | 42.7 |
> | TRACE-uni | 27.5 | 43.9 |
>
> ---

---

> > ### Comment · Reviewer_p7QK · 2024-11-21
> >
> > Thanks for your efforts in addressing my concerns, and I hope you can add these experiments to the final version to make it more comprehensive for readers.

---

> > > ### Author Response · Authors · 2024-11-22
> > > **Reply to Reviewer p7QK**
> > >
> > > Dear Reviewer p7QK,
> > >
> > > Thank you for raising the score! We have incorporated the new results in Appendix B.2 of the revised paper and corrected the illustration about training data as suggested by reviewer YJgy. We appreciate your time and efforts in reviewing our paper.

---

### Official Review · Reviewer_KYwb · 2024-11-03

**Soundness:** 3
**Presentation:** 3
**Contribution:** 2
**Rating:** 8
**Confidence:** 3

**Summary:**

This paper proposes a new method for Video Temporal Grounding (VTG) tasks, named TRACE. TRACE uses a causal event modeling framework to represent videos as a sequence of events with timestamps, salient scores, and textual descriptions. The paper designs a task-interleaved video large language model to address the limitations of traditional video LLMs in handling the inherent structure of videos. The TRACE model utilizes different encoders and decoding heads to process visual frames, timestamps, and text inputs, enabling more effective event sequencing and causal modeling. Experimental results demonstrate that TRACE achieves state-of-the-art zero-shot performance on various VTG tasks and, after fine-tuning, can match the performance of traditional non-generative, task-specific models.

**Strengths:**

1. The paper proposes TRACE, a framework leveraging causal event modeling to generate structured video representations through large language models (LLMs). This approach addresses structural gaps in video data, making it valuable for multi-modal research and practical applications in video analysis.

2. TRACE maximizes the potential of pre-trained LLMs by adopting causal event modeling, which decomposes video inputs into frames and aligns them with textual prompts. The temporal segmentation and alignment methods allow videos to be broken down into events with associated timestamps, salient scores, and captions. This granularity is crucial for precise video event parsing and presents a significant step forward in video understanding with LLMs.

3.TRACE outperforms the existing Video-LLMs on three pivotal Video Temporal Grounding (VTG) benchmarks—Charades-STA, QV Highlights, and YouCook2—underscoring its efficacy and robustness in handling video temporal grounding tasks. This achievement underscores TRACE's ability to accurately capture and model the intricate temporal dynamics across a spectrum of video datasets.

**Weaknesses:**

1.	While causal event modeling is presented as a core contribution of this work, the related work section does not address any prior research on similar methodologies. It would be helpful to clarify whether comparable approaches have been explored in the field of video understanding, or if this approach is entirely novel within this domain. Providing this context could strengthen the argument for the method’s originality and situate it more clearly within existing research.

2.	It is unclear whether compressing visual features to 8 tokens is sufficient for preserving critical information in complex video scenes. The paper does not provide an analysis or experimental results on the trade-off between the number of tokens and model performance, which would be valuable in understanding the potential impact of this compression choice.

3.	There are several grammatical and spelling errors throughout the manuscript, which impact readability and may detract from the paper’s clarity. For example: Line 22: "processes" should be corrected to "process". Line 44-45: The phrase "...which,..." should be rephrased, and "lacks" should be changed to "which lack".

**Questions:**

No additional questions. Please see the "Weaknesses" section for areas needing clarification.

### Recommendations for Improvement:
- **Refine Prompt Design Explanation:** Providing specific strategies or insights on prompt design tailored for VTG tasks would enhance the paper's originality and usefulness for future researchers.

- **Explore Custom Scene Parsing Techniques:** Introducing refined parsing methods could strengthen TRACE's robustness and accuracy in multi-modal alignment.

This structured feedback should provide the authors with a comprehensive view of the strengths and areas for enhancement in their paper on TRACE.

---

> ### Author Response · Authors · 2024-11-17
> **Reply to Reviewer KYwb (1/2)**
>
> Thank you for your valuable suggestions! We have addressed the concerns you raised below.
>
> > While causal event modeling is presented as a core contribution of this work, the related work section does not address any prior research on similar methodologies. It would be helpful to clarify whether comparable approaches have been explored in the field of video understanding, or if this approach is entirely novel within this domain. Providing this context could strengthen the argument for the method’s originality and situate it more clearly within existing research.
> >
>
> Thank you for pointing that out! To the best of our knowledge, **TRACE is the first video-LLM method to model the response of video LLMs as a series of event triplets (timestamps, scores, captions).** To illustrate:
>
> - *Existing video LLMs typically only using text tokens*—either directly representing times/scores by text [1, 2, 3], or adding new tokens to the text vocabulary [4, 5, 6]. However, these methods still generate output in human language, which limits their ability to effectively capture the underlying structure of the video. Furthermore, introducing new tokens into the vocabulary could potentially degrade the pretrained captioning capabilities of the LLMs [6]. *TRACE addresses these challenges by directly modeling video LLM responses as a series of events, using separate encoders and decoders to handle timestamps, scores, and text independently.*
> - *Some non-LLM methods use event-like structures, but they lack causal modeling and are difficult to adapt to video LLMs.* For example, PDVC [7] uses distinct heads to decode timestamps and text, while UniVTG [8] reformats VTG tasks and employs three separate heads. While these methods share some intuitive similarities with the design of TRACE, they are challenging to integrate with LLMs pretrained using causal language modeling. *In contrast, TRACE introduces causal event modeling, enabling easier utilization of pretrained LLMs' reasoning capabilities and knowledge.*
>
> Overall, we believe TRACE is a significant contribution to advancing the field of video LLMs, particularly in addressing the challenges of VTG tasks. Its novel approach could pave the way for future research in effectively integrating video and language models.
>
> > It is unclear whether compressing visual features to 8 tokens is sufficient for preserving critical information in complex video scenes. The paper does not provide an analysis or experimental results on the trade-off between the number of tokens and model performance, which would be valuable in understanding the potential impact of this compression choice.
> >
>
> Thank you for your valuable suggestion! We would like to provide the following clarification
>
> - *We compress the visual tokens to address efficiency and context length limitations.* Since TRACE samples 128 frames, without compression, the ViT would produce over 70K visual tokens. To handle this, we compress the visual tokens to 8 tokens per frame, resulting in a total of 1,792 visual tokens after incorporating the time tokens corresponding to each frame. This compression allows us to effectively handle VTG tasks within the 4K context length limit.
> - *We choose slot-based compression for its lightweight architecture and high performance on VTG tasks.* Introduced by [6], slot-based compression uses only one-third of the parameters of a single cross-attention layer, while outperforming both cross-attention and sampling-based methods on VTG tasks.
> - As per your recommendation, we have conducted ablation studies on the number of tokens per frame. **However, the training will take more than a week to complete. We will post the results here once the ablation training is finished.**

---

> ### Author Response · Authors · 2024-11-17
> **Reply to Reviewer KYwb (2/2)**
>
> > There are several grammatical and spelling errors throughout the manuscript, which impact readability and may detract from the paper’s clarity. For example: Line 22: "processes" should be corrected to "process". Line 44-45: The phrase "...which,..." should be rephrased, and "lacks" should be changed to "which lack".
> >
>
> Thank you for your detailed suggestions! We have corrected the typos in the revised paper, which are highlighted in blue font.
>
> > Refine Prompt Design Explanation
> >
>
> Thank you for the suggestion! We have provided the examples of QA pairs in the Appendix of our submission.
>
> > Explore Custom Scene Parsing Techniques
> >
>
> Thank you for the suggestion! In TRACE, since timestamps, scores, and captions are decoded using separate heads, the parsing process is relatively straightforward. This allows us to independently collect timestamps, scores, and text, and then directly use these collected results for evaluation.
>
> [1] VideoLLaMA 2: Advancing Spatial-Temporal Modeling and Audio Understanding in Video-LLMs. Arxiv 2024.
>
> [2] Vtimellm: Empower llm to grasp video moments. CVPR 2024.
>
> [3] Timechat: A time-sensitive multimodal large language model for long video understanding. CVPR 2024.
>
> [4] Lita: Language instructed temporal-localization assistant. ECCV 2024.
>
> [5] Momentor: Advancing Video Large Language Model with Fine-Grained Temporal Reasoning. ICML 2024.
>
> [6] VTG-LLM: Integrating Timestamp Knowledge into Video LLMs for Enhanced Video Temporal Grounding. Arxiv 2024.
>
> [7] End-to-End Dense Video Captioning with Parallel Decoding. ICCV 2021.
>
> [8] UniVTG: Towards Unified Video-Language Temporal Grounding. ICCV 2023.

---

> > ### Author Response · Authors · 2024-11-19
> > **Results on number of slots per frame**
> >
> > Dear Reviewer KYwb,
> >
> > We have finished the experiments that compressing each frame into 16 tokens. Due to time constraints, we adopted the same settings as in Table 3, using VTG-IT only in Stage 2 and sampling 64 frames. Our findings are as follows: Increasing the number of slots per token significantly enhances TRACE's performance. Therefore, if computational or efficiency constraints are not a concern, we recommend using a larger number of slots per frame.
> >
> >
> >
> > | Youcook2 | Frame Num | Slot Num per Frame | SODA_c | CIDEr |  F1 Score |
> > | --- | --- | --- | --- | --- | --- |
> > |  | 64 | 8 | 1.9 | 6.9 | 21.4 |
> > |  | 64 | 16 | 2.1 | 7.3 | 22.1 |
> >
> > | Charades-STA | Frame Num | Slot Num per Frame | R@1$_{IOU=0.5}$ | R@1$_{IOU=0.7}$ |
> > | --- | --- | --- | --- | --- |
> > |  | 64 | 8 | 37.0 | 17.0 |
> > |  | 64 | 16 | 41.9 | 20.1 |
> >
> > We hope our responses have sufficiently addressed your concerns. Please do not hesitate to reach out if you have any further questions. Thank you again for your time and effort in reviewing our paper.

---

> > > ### Comment · Reviewer_KYwb · 2024-11-26
> > > **Reply to the authors' response**
> > >
> > > I have no further questions and tend to increase the rating.

---

> > > > ### Author Response · Authors · 2024-11-26
> > > >
> > > > Dear reviewer KYwb,
> > > >
> > > > Thank you for raising the score! Your detailed review has been invaluable in helping us improve the quality of the paper.

---

### Meta-Review · Area_Chair_QWCP · 2024-12-16

**Metareview:**

The paper receives 3 positive and 1 negative ratings after rebuttal, with 3 upgraded scores. Initially, the reviewers had several concerns about some technical clarity, motivations of using timestamps and scores, more contexts with the relevant work, more analysis on model parameters, extending to other video tasks, comparisons with multimodal LLMs (e.g., VideoLLama2). In the post-rebuttal discussion period, three reviewers were satisfactory with the authors' comments and raised the rating. After taking a close look at the paper, rebuttal, and discussions, the AC agrees with reviewers' feedback of the proposed method being effective and significant as a LLM foundation for video temporal grounding. Therefore, the AC recommends the acceptance rating.

**Additional Comments On Reviewer Discussion:**

In the rebuttal, most critical concerns from the reviewer KYwb, p7QK, and YJgy, about technical clarity and more experimental results (e.g., comparisons with other video LLMs, other video tasks) are well received by the reviewers. Moreover, for the reviewer MMw2 who still provides the negative rating, the main concerns are on the usage of timestamps/scores and comparisons with Qwen2-VL. The AC took a close look at the rebuttal, discussions, and responses, in which the AC finds that the raised issues are addressed well by the authors in the rebuttal.

---

### Decision · Program_Chairs · 2025-01-22

Accept (Poster)